# Ground- and ship-based microwave radiometer measurements during EUREC4A

Sabrina Schnitt[1], Andreas Foth[2], Heike Kalesse-Los[2], Mario Mech[1], Claudia Acquistapace[1], Friedhelm Jansen[3], Ulrich Löhnert[1], Bernhard Pospichal[1], Johannes Röttenbacher[2], Susanne Crewell[1], and Bjorn Stevens[3]

[1]Institute for Geophysics and Meteorology, University of Cologne, Cologne, Germany
[2]Leipzig Institute for Meteorology (LIM), Leipzig University, Leipzig, Germany
[3]Max Planck Institute for Meteorology, Hamburg, Germany

**Correspondence:** Sabrina Schnitt (s.schnitt@uni-koeln.de)

**Abstract.**

During the EUREC[4]A field study, microwave radiometric measurements were performed at Barbados Cloud Observatory (BCO) and aboard the RV Meteor and RV Maria S Merian in the down-stream winter trades of the North Atlantic. We present retrieved Integrated Water Vapor (IWV), Liquid Water Path (LWP) and temperature and humidity profiles as a unified, quality-controlled, multi-site dataset on a three second temporal resolution for a core period between January 19, 2020 and February 14, 2020 in which all instruments were operational. Multi-channel radiometric measurements were performed at BCO and aboard the RV Meteor between 22 and 31 GHz (K-band), and 51 to 58 GHz (V-band). Combined radar-radiometer measurements of a W-band Doppler radar with a single-channel radiometer instrument were conducted at 89 GHz onboard the RV Meteor and RV Maria S Merian. We present a novel retrieval method to retrieve LWP from single-channel 89 GHz measurements, evaluate retrieved quantities with independent measurements and analyze retrieval uncertainties by site and instrument inter-comparison. Mean IWV conditions of $31.8\,\mathrm{kg\,m^{-2}}$ match independent radiosoundings at BCO with a root-mean-square difference of $1.1\,\mathrm{kg\,m^{-2}}$. Mean LWP conditions in confident liquid cloudy, non-precipitating conditions ranged between $63.1\,\mathrm{g\,m^{-2}}$ at BCO to $46.8\,\mathrm{g\,m^{-2}}$ aboard the RV Maria S Merian. Aboard the ships, $90\,\%$ of LWP was below $120\,\mathrm{g\,m^{-2}}$ with a $30\,\%$ uncertainty for LWP of $50\,\mathrm{g\,m^{-2}}$. Up to $20\,\%$ of confident liquid cloudy profiles ranged below the LWP detection limit due to optically thin clouds.

The data set comprises of processed raw-data (Level 1), full quality-controlled post-processed instrument data (Level 2), a unified temporal resolution (Level 3), and a ready-to-use multi-site time series of IWV and LWP (Level 4), available to the public via AERIS (https://doi.org/10.25326/454#v2.0, Schnitt et al., 2023a). The data set complements the airborne LWP measurements conducted during EUREC[4]A and provides a unique benchmark tool for satellite evaluation and model-observation studies.

# 1 Introduction

The subtropical oceans are ubiquitously covered by shallow trade-wind cumulus clouds. While small in individual size and height, cloud fields are large in their extent, which makes them important for the radiative budget through the short-wave reflected radiation which is directly related to the liquid water amount and distribution in the cloud. Large inter-model spreads of climate sensitivity are thought to be related to the representation of these clouds in current climate models (Bony et al., 2015; Dufresne and Bony, 2008; Vial et al., 2013; Zelinka et al., 2020; Jahangir et al., 2021) and their potential role in mediating the long-wave radiative response to warming (Stevens and Kluft, 2023). Open questions include the interaction of these clouds with their environment, and their coupling to circulation and convection (Bony et al., 2017).

In order to elucidate the underlying processes of the interactions, high quality and fine resolution observations were gathered during the EUREC[4]A field study in January and February 2020 (Stevens et al., 2021) over the Tropical Atlantic windward and in the close vicinity of Barbados. A range of complementary atmospheric and oceanic measurements were performed by four different research aircraft (Konow et al., 2021; Bony et al., 2022; Pincus et al., 2021), and by ground- and ship-based observations (e.g. Acquistapace et al., 2022; Kalesse-Los et al., 2023). Microwave radiometric measurements were conducted at Barbados Cloud Observatory (BCO, Stevens et al., 2016), as well as aboard the RV Meteor (later referred to as Meteor) and the RV Maria S Merian (referred to as Merian). These measurements manifest an important contribution to the overall data set as they quantify cloud liquid water and water vapor amount statistically at high temporal resolution. Here, we present the data set of Integrated Water Vapor (IWV), Liquid Water Path (LWP) as well as profiles of temperature $T$ and absolute humidity $\rho_v$ retrieved from the measurements at BCO, and aboard the Meteor and Merian.

Passive microwave radiometry is widely in use from satellite, airborne platforms such as the High Altitude LOng range (HALO) aircraft (Mech et al., 2014; Stevens et al., 2019), research vessels like the RV Polarstern (Walbröl et al., 2022), and ground-based supersites like the Atmospheric Radiation Measurement (ARM) program (Stokes and Schwartz, 1994) or Cloud-Net (Illingworth et al., 2007) to continuously measure IWV and LWP. LWP conditions in the Northern Atlantic winter trades have been previously measured during the RICO (Rain in shallow Cumulus over the Ocean Rauber et al., 2007) campaign. During the Next Generation Aircraft Remote Sensing for Validation Studies (NARVAL) I and II campaigns (Stevens et al., 2019; Jacob et al., 2019; Schnitt et al., 2017), airborne microwave radiometric measurements were performed by the HALO microwave package HAMP (HAMP, Mech et al., 2014), providing benchmark observations to elucidate cloud and precipitation properties in storm-resolving models (Jacob et al., 2020). During EUREC[4]A, airborne measurements were again performed by the HALO-HAMP as described in Konow et al. (2021) and available in Jacob (2021). Space-borne observations of LWP in warm oceanic clouds reveal large biases depending on the used sensor and retrieval approach (Seethala and Horvath, 2010; Elsaesser et al., 2017). The here presented ground- and ship-based MWR measurements therefore provide an important high-resolution data set of IWV and LWP to evaluate air- or space-borne retrievals and to benchmark existing and future modelling experiments.

As opposed to remote sensing in the visible or infrared parts of the spectrum, passive microwave radiometer (MWR) measurements are sensitive to the full vertical column as clouds are semi-transparent in the microwave frequencies. Water vapor,

oxygen, and liquid water emit at characteristic frequencies. Emissions can be measured as brightness temperatures $T_B$ following Planck's law. While water vapor and oxygen emit in distinct absorption bands in the K- and G-band (around 22.2 GHz and 183.3 GHz, respectively), and V- and F-band (60.0 GHz and 118.8 GHz, respectively), liquid water emissions increase with increasing frequency (Ulaby, 2014). Therefore, channels in the water vapor sensitive K-band need to be paired with measurements from a window channel around 31.4 GHz or 90 GHz to allow a simultaneous retrieval of IWV and LWP (Westwater, 1978; Löhnert and Crewell, 2003). Single-channel measurements around 90 GHz provide higher sensitivity to LWP, but require knowledge of IWV to solve the underdetermined inversion problem (e.g. Westwater et al., 2001; Billault-Roux and Berne, 2021). Absolute humidity profiles with limited vertical resolution (Löhnert et al., 2009) can be derived if multiple channels are located along the wing of the 22.24 or 183 GHz line. Temperature profiles of better than 500 m vertical resolution can be obtained from the oxygen absorption complex around 50 GHz. A higher resolution can be achieved by scanning at different elevation angles (Crewell and Löhnert, 2007). A scattering contribution to the measured $T_B$ only occurs if ice is present in clouds for frequencies above 90 GHz (e.g. Weng et al., 2003).

The $T_B$ measured by the Humidity and Temperature PROfiler HATPRO (Rose et al., 2005), a fourteen-channel state-of-the-art microwave radiometer, allow the retrieval of IWV, LWP, as well as temperature and humidity profiles based on statistical regression techniques (Löhnert and Crewell, 2003), physical retrievals (Turner et al., 2007a; Maahn et al., 2020) or neural networks (Cadeddu et al., 2009; Jacob et al., 2019). Measurements can only be obtained in non-precipitating conditions as a wet radome causes non-atmospheric liquid emissions. A HATPRO is permanently installed at BCO (Stevens et al., 2016), here referred to as BCOHAT. During EUREC[4]A, BCOHAT measurements were complemented by HATPRO measurements aboard the Meteor performed by the Leipzig Institute for Meteorology (LIM), here referred to as LIMHAT. Aboard the Meteor, a 94 GHz cloud radar (Küchler et al., 2017) was installed, equipped with a passive radiometer channel measuring $T_B$ at 89 GHz (Kalesse-Los et al., 2023), here referred to as LIMRAD. A similar instrument was operated aboard the Merian (Acquistapace et al., 2022), here referred to as MSMRAD. As water vapor and liquid water both contribute to the single-channel $T_B$, we retrieve IWV from LIMRAD and MSMRAD only in clear-sky conditions. We use a novel retrieval method to derive cloudy LWP from the brightness temperature difference between cloudy and clear-sky $T_B$ rather than from absolute $T_B$ measurements.

This paper describes the network of continuous ground- and ship-based microwave radiometer measurements in a core period of January 19, 2020, until February 14, 2020, during which all four instruments were operational. We document setup and installation of the instruments (Sec 2), introduce the retrieval methods (Sec 3) and describe precipitation and cloud masking as well as data processing (Sec 4). We use independent measurements to derive and evaluate the retrieved IWV (Sec 5) and analyze LWP conditions and uncertainties (Sec 6). Retrieved temperature and humidity profiles are discussed in Sec 7. We conclude the paper in Sec 8 by summarizing and highlighting further scientific applications for this data set.

## 2 MWR network

During EUREC[4]A, passive radiometer measurements were performed from BCO, the Meteor, and the Merian. The following subsections describe the installation details of the instruments at each site, respectively. Instruments' details and retrieved quan-

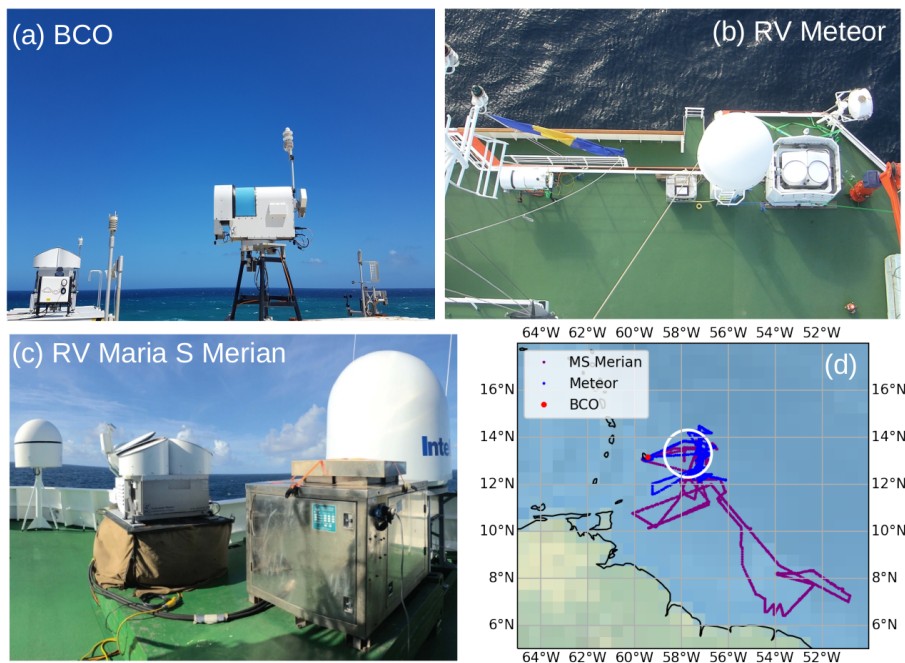

**Figure 1.** Installation of (a) MWR BCOHAT at BCO, (b) MWR LIMHAT and cloud radar LIMRAD aboard the Meteor, (c) cloud radar MSMRAD aboard the Merian, and (d) map of operations with BCO (red), and Meteor (blue) and Merian (purple) ship tracks, including the circle flown by the HALO aircraft (white) for orientation.

tities are summarized in Table 1. Installation and map of operations are shown in Fig 1. Microwave radiometer measurements were not performed aboard the RV Ronald H Brown. While an inter-platform comparison is generally performed statistically, two distinct periods of measurement allow a direct comparison of the ship-based measurements. On January 19, 2020 between 00 and 12 UTC, both RVs were steaming next to one another from -58 °W to -57.3 °W between 13.8 and 13.75 °N. On February 07, 2020, the ships were collocated at -57.2 °W and 12.4 °N between 11 and 18 UTC.

## 2.1 BCO

The RPG-HATPRO Generation 5 multi-channel microwave radiometer BCOHAT is continuously installed on top of a container at 25 m asl in proximity to the island shore (see Fig 1a and Stevens et al., 2016). An absolute calibration with liquid nitrogen was performed before the start of the EUREC⁴A operations on January 14, 2020. BCOHAT measured according to the following regularly-occurring scan strategy: azimuth scans at 30° elevation angle were performed for the duration of 5 minutes every 40 minutes, followed by an elevation scan covering 11 elevation angles (90. , 30, 19.2, 14.4, 11.4, 8.4, 6.6, 5.4, 4.8, 4.2°) at 0° azimuth position (later referred to as elevation scan, Crewell and Löhnert (2007)). Zenith measurements are performed for 15 minutes at a temporal resolution of 2 seconds. Due to technical difficulties with the scanning unit, the scanning strategy changed after February 1, 2020: azimuth scans were not performed, and operations were limited to zenith mode with elevation scans

**Table 1.** Overview of passive microwave measurements performed during EUREC[4]A at BCO, aboard the Meteor and the Merian. Measured quantities, retrieved variables, each instrument's scan strategy as well as the covered time periods are given.

| | BCO | Meteor | | Merian |
|---|---|---|---|---|
| instrument | **BCOHAT** (Rose et al., 2005) | **LIMHAT** | **LIMRAD** (Kalesse-Los et al., 2023) | **MSMRAD** (Acquistapace et al., 2022) |
| $T_B$ measured at | 22.24 - 31.4 GHz (7 channels) 51 - 58 GHz (7 channels) | same as BCOHAT | 89.0 | 89.0 |
| retrieved quantities | IWV, LWP T, $\rho_v$ profiles | IWV, LWP T, $\rho_v$ profiles | clear-sky IWV, LWP | clear-sky IWV, LWP |
| scan strategy | zenith elevation scan every 15 min | zenith unstabilized elevation scan full hour | zenith, stabilized | zenith, stabilized |
| time coverage | January 01 - February 14, 2020 | January 15 - February 19, 2020 | January 17 - February 19, 2020 | January 16 - February 19, 2020 |

available every 15 minutes. These technical difficulties also affected the associated BCOHAT weather station. From January 26, 2020, onwards, data from the adjacent BCO weather station was used instead to flag measurements for precipitation (https://doi.org/10.25326/54). No measurements were performed between January 29 and 31, 2020, due to maintenance on the instrument. A blowing unit was operational to mitigate the deposition of rain on the radomes during and after precipitation events.

## 2.2 Meteor

Aboard the Meteor, the Leipzig Institute for Meteorology (LIM) of Leipzig University operated an MWR type RPG-HATPRO Generation 5 (here referred to as LIMHAT) and a radar-radiometer system of type RPG-FMCW-94 dual polarization (DP) operating actively in the W-band (94 GHz) and containing a passive radiometer channel at 89 GHz (Küchler et al., 2017; Kalesse-Los et al., 2023, here referred to as LIMRAD). Both instruments were placed 4.5 m apart on the navigation deck of the ship at 15.8 m above sea Level to avoid sea spray. LIMHAT operated at a temporal resolution of 1 s in zenith mode. Elevation scans, as done by BCOHAT, were performed by LIMHAT every full hour. An absolute calibration with liquid nitrogen was performed on January 15, 2020.

LIMRAD was operated with two different radar settings as specified in Kalesse-Los et al. (2023). Between January 17 and 29, 2020, and Jan 31 and February 28, 2020, the temporal resolution of LIMRAD was 2.9 s and 1.6 s, at a vertical resolution between 22 and 42 m, respectively. Radar absolute calibration was performed on January 16, 2020. Data gaps exist between January 27 and 31, 2020 when different radar chirp table settings were tested, and on February 03, 2020, when all instruments had to be turned off while the Meteor was near Trinidad. As explained in Kalesse-Los et al. (2023), LIMRAD was operated in a novel passive horizontal stabilization system (two-axle cardanic mount) to assure zenith-pointing of the instrument. Stabilization is required to eliminate the effect of horizontal wind on the radar Doppler velocities. Means and standard deviations of absolute

values of radar attitude measurements amounted to $0.36°+-0.31°$. It should be noted that since LIMRAD was operated in a horizontal stabilization platform while LIMHAT was not, the exact (near-zenith) viewing direction of both instruments was not always the same. This effect should be negligible for retrieved IWV and LWP, however, as the larger opening angle of the LIMHAT (half-power beamwidth HPBW = $3.5°$) covered the LIMRAD column (HPBW = $0.5°$) even in events of slight mis-pointing.

## 2.3 Merian

Aboard the Merian, the Institute for Geophysics and Meteorology of the University of Cologne operated a radar-radiometer system of the type RPG-FMCW-94 dual polarization (DP) which measures in the W-band (94 GHz) and includes a passive radiometer channel at 89 GHz (Küchler et al., 2017, here referred to as MSMRAD). MSMRAD is of the same type as LIMRAD. The system was positioned on an active stabilization platform from the US-Atmospheric Radiation Measurement (ARM) program Mobile Facility 2 which keeps the radar in zenith position by adapting the table surface position to compensate for ship motions (for more information see Acquistapace et al., 2022). As for LIMRAD, stabilization helps eliminate the effect of horizontal wind and ship roll and pitch tilting from the radar Doppler velocities. MSMRAD was operated with three chirp programs, established after initial testing, and worked for the entire campaign. The chirp programs had 0.846, 0.786, and 1.124 s integration time, respectively, with vertical resolution of 7 m in the boundary layer and 30 m in the free troposphere, resulting in brightness temperatures at 3 second temporal resolution (see Table 2 in Acquistapace et al., 2022). The data browser (https://bit.ly/3ZcAusN) displays availability and observational quality for every day of the entire campaign.

## 3 Retrievals

This section presents the statistical retrieval methods applied to the HATPRO and single-channel 89 GHz measurements. IWV, LWP, temperature and humidity profiles are retrieved from BCOHAT and LIMHAT, while LWP and clear-sky IWV are retrieved from LIMRAD and MSMRAD. We make use of a state-of-the-art retrieval using $T_{\mathrm{B}}$ from all fourteen HATPRO channels (Sec. 3.1). In order to disentangle water vapor and liquid contributions in the 89 GHz $T_{\mathrm{B}}$ retrieval, we retrieve IWV in clear-sky conditions only, and present a novel retrieval method to derive single-channel LWP by retrieving from a $T_{\mathrm{B}}$ difference of cloudy $T_{\mathrm{B}}$ and closest clear-sky $T_{\mathrm{B}}$ (Sec. 3.2).

### 3.1 multi-channel retrieval: BCOHAT and LIMHAT

IWV, LWP as well as coarse temperature and humidity profiles are retrieved from all fourteen $T_{\mathrm{B}}$ measurements by applying the statistical quadratic regression retrieval equation (see Eq. 1 with $k$ indicating the number of channel and $var$ indicating the retrieved variable). The seven K-band channels (22-31 GHz, channel 1-7) provide information for IWV, LWP and the absolute humidity profiles, while the seven V-band channels (51-58 GHz, channels 8-14) are used for temperature profiling. The IDL software *MWR PRO* was used to process the data (Löhnert, 2023).

$$\text{var} = c_0 + \sum_{k=0}^{N=6} c_{1,k} \cdot T_{\text{B},k} + c_{2,k} \cdot T_{\text{B},k}^2 \tag{1}$$

The coefficients $c_0$, $c_1$ and $c_2$ are derived from a climatological training data set comprising of 10,871 daily radiosoundings launched from 1990 until 2018 from Grantley Adams International Airport (GAIA, station ID 78954 TBPB) in close vicinity to BCO. Sounding measurements were obtained from http://weather.uwyo.edu/upperair/sounding.html. During EUREC$^4$A radiosoundings of the type GRAW DFM-09 were used (Bock et al., 2021).

Following the approach by Löhnert and Crewell (2003) and, more recently, by Walbröl et al. (2022), we use a radiative transfer model to link atmospheric conditions with $T_\text{B}$. In the model, gas absorption is calculated according to Rosenkranz (1998) with modifications in the water vapor continuum (Turner et al., 2009) and 22 GHz line (Liljegren et al., 2005). Liquid cloud absorption is calculated following Mätzler et al. (2006). A liquid water cloud was modeled using a modified adiabatic liquid water content approach following Karstens et al. (1994) in vertical levels where radiosounding relative humidity exceeded 95 %. To imitate the instrument's noise, a random noise factor was added to the simulated $T_\text{B}$ taken as a random sample from a Gaussian distribution with standard deviation of 0.4 K (Maschwitz et al., 2013). For the temperature retrieval, only a linear regression was used as in Walbröl et al. (2022). To derive temperature profiles from the elevation scans, coefficients $c_1$ and $c_2$ were calculated by adjusting the angle for which radiative transfer was performed. Theoretical LWP uncertainty scales with retrieved LWP, and is further discussed in Sec 6.2.

To further improve the stand-alone LWP retrieval, a clear-sky offset correction method is applied to the retrieved LWP (van Meijgaard and Crewell, 2005; Ebell et al., 2017). The correction scheme identifies a liquid-free condition if the standard deviation of LWP in a running 2-minute window, as well as the previous and subsequent 2-minute window, is below $2.5\,\text{g}\,\text{m}^{-2}$. The median LWP during the identified 2-minute clear-sky period is subsequently subtracted from all following LWP measurements until the next clear-sky period. Note that due to the statistical retrieval approach, negative (unphysical) LWP values can occur. Remaining few negative LWP values are not set to zero to keep for statistical noise evaluation and to avoid biasing the overall statistical distribution of LWP. That way, the clear-sky LWP noise can be estimated by analyzing the LWP distribution in independently identified clear-sky periods as presented in Sec 6.2.

## 3.2 single-channel retrieval: LIMRAD and MSMRAD

As opposed to a multi-channel LWP retrieval, the retrieval of LWP from a single channel is underdetermined as both water vapor and liquid water contribute to the measured $T_\text{B}$ (e.g. Westwater et al., 2001; Billault-Roux and Berne, 2021). In order to extract the LWP signal in $T_\text{B}$ at 89 GHz, we present a novel retrieval approach based on the difference in brightness temperature $\Delta T_\text{B}$ between cloudy-sky $T_\text{B}$ and the closest clear-sky $T_{\text{B},0}$. $\Delta T_\text{B}$ is used in a third-order regression (Eq. 2) to estimate LWP.

$$\text{LWP} = a \cdot \Delta T_\text{B} + b \cdot \Delta T_\text{B}^2 + c \cdot \Delta T_\text{B}^3 \quad \text{with} \quad \Delta T_\text{B} = T_\text{B} - T_{\text{B},0} \tag{2}$$

Instrument biases are reduced by using the difference in brightness temperatures, so that the unbiased portion of the signal from LWP remains. The clear-sky brightness temperature is obtained by selecting profiles not showing any radar reflectivity through the cloud mask, excluding measurements up to 5 min after rain events to avoid biases due to wet radome conditions. The unknown coefficients of the regression ($a$, $b$, and $c$) are derived from a training data set compiled from artificial LWPs and simulated brightness temperatures calculated with the forward model operator Passive and Active Microwave TRAnsfer model (PAMTRA; Mech et al., 2020). Atmospheric profiles were constructed from 401 radiosondes launched on the respective research vessels (Merian 182, Meteor 219) and artificial clouds between 0 and 5 km with LWPs up to 1 kgm$^{-2}$. To retrieve LWP from the measured $\Delta T_{\mathrm{B}}$ in non-precipitating conditions, the coefficients derived for the closest radiosounding were applied following Eq. 2 to $\Delta T_{\mathrm{B}}$ which was in turn noised by a random number of a Gaussian distribution with width of 0.5K.

IWV is retrieved from the single channel $T_{\mathrm{B}}$ measurements only in clear-sky conditions as emissions then are dominated by water vapor. A quadratic regression is applied as in Eq. 1, weighed by variability of $T_{\mathrm{B}}$ around the radiosonde launch. By applying a weight to the regression, mis-identified clear-sky radiosoundings are excluded from the training. 120 and 65 clear-sky radiosoundings were identified aboard the Meteor and Merian, respectively, by applying a 98 % relative humidity threshold, and were used to derive the coefficients linking $T_{\mathrm{B}}$ and IWV. The coefficients were then applied to the measured $T_{\mathrm{B}}$ in clear-sky conditions as detected by the cloud masking algorithm presented in the following Section.

## 4 Masking and Data Processing

This Section describes the processing of the data set as available on AERIS, https://doi.org/10.25326/454#v2.0 (Schnitt et al., 2023a). Section 4.1 describes the precipitation and cloud masking and Sec. 4.2 summarizes the processing of the measurements from Level 1 to Level 4.

### 4.1 Precipitation and Cloud Masking

Ground-based passive microwave radiometer measurements are not reliable during precipitation events due to additional liquid water emissions on the radome contributing to the column emissions. Flagging precipitation is, thus, crucial to guarantee a high-quality retrievals. The HATPRO precipitation mask is set to True when precipitation was detected by the internal HATPRO or an adjacent weather station. Cloud radar measurements are added to the stand-alone precipitation flagging to improve the precipitation detection. At BCO, Ka-band (35 GHz) zenith-pointing radar measurements (Hirsch, 2022) are used. Aboard the ships, measurements of the LIMRAD and MSMRAD cloud radar operating at W-band (94 GHz) are added. Precipitation is flagged if any reflectivity above -50 dBZ was recorded below 350 m. This reflectivity threshold was chosen according to Klingebiel et al. (2019) to exclude sea salt aerosols from being mis-flagged as precipitation. Aboard the ships, precipitation was also flagged if reflectivity exceeded 0 dBZ anywhere in the column (Kalesse-Los et al., 2023) or if a rain rate was derived by the radar.

Independent cloud masking was performed using the adjacent radar and, at BCO and aboard the Meteor, ceilometer measurements from a Jenoptik/Lufft CHM15k Nimbus ceilometer, respectively. Ceilometer measurements are identified as cloudy

if a cloud base height above 100 m is derived by the internal instrument software. If no valid cloud base height is derived, the scene is treated as clear. At BCO and aboard the Merian, radar measurements indicate cloudy conditions if a reflectivity of more than -50 dBZ is recorded in more than two range gates above 300 m. The reflectivity threshold was carefully chosen to exclude occurring sea-spray from being flagged as cloudy (Klingebiel et al., 2019). Due to the different radar chirp settings (see Sec. 2.2) and resulting radar sensitivities, a threshold of -40 dBZ was applied to the LIMRAD measurements to optimally exclude sea spray and clutter. An additional liquid cloud mask is derived by enforcing that reflectivity above the respective threshold only occurred between 300 and 4000 m. Clear-sky is identified if reflectivity is nan in all range bins. Due to MSM-RAD's maximal range of 10 km, high occuring cirrus clouds might not be detected and could be mis-flagged as clear conditions aboard the Merian.

In the presented analyses, the individual cloud masks are combined to a joint cloud mask as follows: *clear* conditions prevail if both ceilometer and radar flags are clear; *probably cloudy* conditions prevail if either ceilometer or radar sensed a cloud; and *confident cloudy* scenes refer to measurements in which both radar and ceilometer sensed a cloud. Making use of the additional liquid cloud flag allows to additionally derive *probably liquid cloudy* and *confident liquid cloudy* conditions to exclude scattering from ice in the LWP statistics. Probably cloudy occurrences are mainly due to sensor beam mismatch, platform motions, or sensitivity differences between the ceilometer and radar as outlined in Konow et al. (2021). Aboard the Merian, scenes were classified as clear or confident cloudy based solely on MSMRAD radar observations.

Missing data of ceilometer or radar led to discarding of 3.7 %, 20 % and 8.2 % of all measurements as a cloud mask could not be determined at BCO, Meteor, Merian, respectively. The comparatively higher percentage aboard the Meteor is dominated by data availability of LIMRAD. For presented analyses, we additionally demand a valid IWV and LWP, as well as a valid cloud mask for a measurement to be considered, thus excluding scenes affected by precipitation or instrument measurement or retrieval quality. This reduces the availability of valid measurements to 50.5, 66.8, 69.5, and 83.1 % of all 3 s measurements in the core period, dominated by instrument availability as shown in Fig 2.

Table 2 summarizes the respective cloud cover fractions of clear, probably (liquid) cloudy and (liquid) cloudy scenes relative to this subsample for all instruments. Clear-sky fraction is highest aboard the Merian, and lowest at BCO. We relate the highest clear-sky fraction of 75.2 % aboard the Merian to the missing ceilometer and reduced sensitivity of the radar to optically thin and geometrically small clouds (Mieslinger et al., 2022). Compared to the airborne cloud cover products presented in Konow et al. (2021), the here presented ground-based derived confident cloudy cloud cover estimates are closest to the airborne lidar-derived cloud cover of 34 %. Differences arise due to the fact that airborne operation was limited to selected days and daytime, and that airborne horizontal resolution is lower than when measured from ground. Here presented cloud cover matches the cloud cover observed at BCO from two years of measurements (Nuijens et al., 2014). More than 80 % of detected clouds are classified as liquid.

## 4.2 Overview of Processing Levels

**Level 1** Level 1 files are provided for each instrument and include the unfiltered instrument output on original time resolution. HATPRO measurements were processed by the MWR PRO software (see Sec 3.1), providing one daily file for IWV, LWP, T

**Table 2.** Cloud Mask Characteristics at BCO, aboard the Meteor and Merian. Scenes are *clear* if both ceilometer and radar sensed clear-sky; *probably cloudy* if either detected a cloud; and *confident cloudy* if both radar and ceilometer detected clouds. Fractions for respective liquid cloud occurrence are given in parenthesis. Percentages are relative to total number of non-precipitating measurement points with valid LWP and cloud mask. Liquid fraction refers percentage of liquid clouds of all clouds.

| site | clear | probably cloudy (liquid) | confident cloudy (liquid) | liquid fraction |
|------|-------|--------------------------|---------------------------|-----------------|
|      | %     | %                        | %                         | %               |
| BCOHAT | 48.6 | 11.0 (13.5) | 40.5 (33.5) | 82.6 |
| LIMHAT | 59.0 | 19.0 (3.2)  | 22.1 (19.3) | 87.7 |
| LIMRAD | 61.1 | 16.3 (1.5)  | 22.6 (21.5) | 95.4 |
| MSMRAD | 75.2 | 0.0 (0.0)   | 24.8 (21.0) | 84.5 |

and q retrieval as well as for the $T_B$ measurements. The LIMHAT data set is also available in Kalesse-Los et al. (2020). The HATPRO quality flags include flags for visual inspection, sun influence in measurement beam, and a $T_B$ threshold indicating poor measurement quality. For the W-band measurements, one file per day is produced by the manufacturer's software (see Acquistapace et al., 2022; Kalesse-Los et al., 2023).


**Level 2** One Level 2 file is provided per instrument, concatenating the daily Level 1 HATPRO and hourly W-band files, respectively, in one single file. Measurements and retrieval products are given in the original instrument's time resolution. LWP is clear-sky corrected as described in Sec 3.1. The provided HATPRO quality mask indicates poor measurement and retrieval quality, respectively, combining single flags from Level 1 files in one flag. Poor measurement quality is flagged if any of the Level 1 quality flags is True, and as identified manually due to maintenance on the instruments (see Sec 2). An additional check is performed by simulating $T_B$ for each channel individually based on $T_B$ observations of all other channels. If the difference between simulated and observed $T_B$ is above a certain threshold, the spectrum is considered as unphysical and flagged. These unphysical spectra can be caused by rain, wet radome, or other external sources (such as radio-frequency interference, sun in beam, etc.). Threshold values were determined empirically, and are as follows: at K-band the sum of the absolute differences between channels 2 through 7 is larger than 3 K; at V-band the sum of the absolute differences between all channels is larger than 7 K. Poor retrieval quality is flagged for IWV, LWP, temperature, and humidity independently. In addition to the information given by the instrument's housekeeping data, IWV values larger than $60\,\mathrm{kg\,m^{-2}}$, and LWP values larger than $1000\,\mathrm{g\,m^{-2}}$ are flagged to additionally exclude poor retrieval quality or ice scattering impacts leading to errroneously high retrieval results (see e.g. Jacob et al. (2019)). LWP clipping amounts to 4.3 % (BCOHAT), 1.5 % (LIMHAT), 2.2 % (LIMRAD), 1.5 % (MSMRAD) of all available retrieved LWP. Precipitation was flagged as outlined in the previous Section. As described in Sec 2, the HATPRO instruments performed different measurement strategies deviating from pure zenith measurements. A position mask included in Level 2 data indicates zenith measurement, azimuth or elevation scan measurement. IWV was derived from single-channel 89 GHz measurements in clear-sky conditions as identified by LIMRAD and MSMRAD measurements (see previous Section).




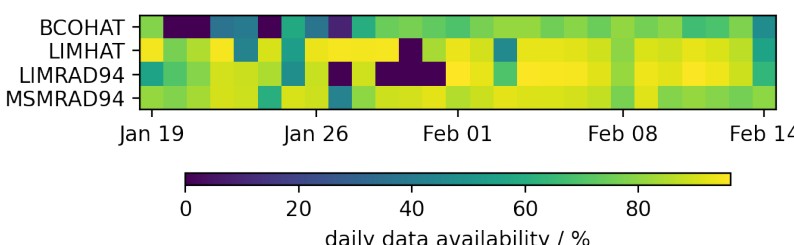

**Figure 2.** Timeline of measurement availability (color-coded, in percent) in the identified core period between January 19 and February 14, 2020, for each instrument. Percentages are calculated with respect to the optimal expected number of measurements on the 3 s temporal resolution grid.

**Level 3** One Level 3 file is provided for each site, combining all available radiometer and single-channel measurements on a mutual 3 s time grid to facilitate inter-platform comparison. A core measurement period was defined ranging from January 19 until February 14, 2020, during which all instruments were operational. As illustrated in Fig 2, certain days did not contain measurements due to maintenance, and precipitation reduced the amount of available measurements. All following analyses, if not indicated differently, are based on the Level 3 data set.

Mean characteristics of the core period are summarized in Table 3. At BCO, aboard the Meteor, and the Merian, respectively, 9.1, 10.7 and 14.6 % of valid precipitation mask time steps were flagged as precipitating at ground. Scenes are flagged confident liquid cloudy in 33.5, 19.3, and 21.0 % of all valid measurements at BCO, Meteor and Merian, and are characterized by a mean LWP of 63.1, 62.5 and 46.8 $\text{g m}^{-2}$, respectively.

**Level 4** Quality-controlled time series of IWV, LWP, precipitation and cloud mask are given in one file for all three sites. Level 4 estimates are based on BCOHAT, LIMHAT, and MSMRAD retrieved IWV and LWP. Additionally, different files are provided for timelines of IWV and LWP sampled to different temporal resolution: 3 seconds (orginal), 1 minute, 30 minutes, 1,3, 6, 12 hours, and daily. The 6-hourly timeline of IWV, LWP, 6-hourly variability of LWP, as well as daily precipitation fraction are illustrated in Fig 3. Spikes in LWP and LWP variability are related to unidentified precipitation events. While IWV varies little when sampled daily, longer sampling times smooth the LWP distribution.

**Table 3.** Characteristics of clouds, water vapor, precipitation and liquid cloud occurrence at each site. Precipitation and liquid cloud cover are calculated as temporal fraction of all valid measurements within the core period. Mean LWP is calculated for confident liquid cloudy scenes.

| site | mean IWV $\mathrm{kgm^{-2}}$ | confident liquid cloudiness % | mean LWP $\mathrm{gm^{-2}}$ | precip fraction % |
|---|---|---|---|---|
| BCO | 31.8 | 33.5 | 63.1 | 9.1 |
| Meteor | 30.3 | 19.3 | 62.5 | 10.7 |
| Merian | 33.3 | 21.0 | 46.8 | 14.6 |

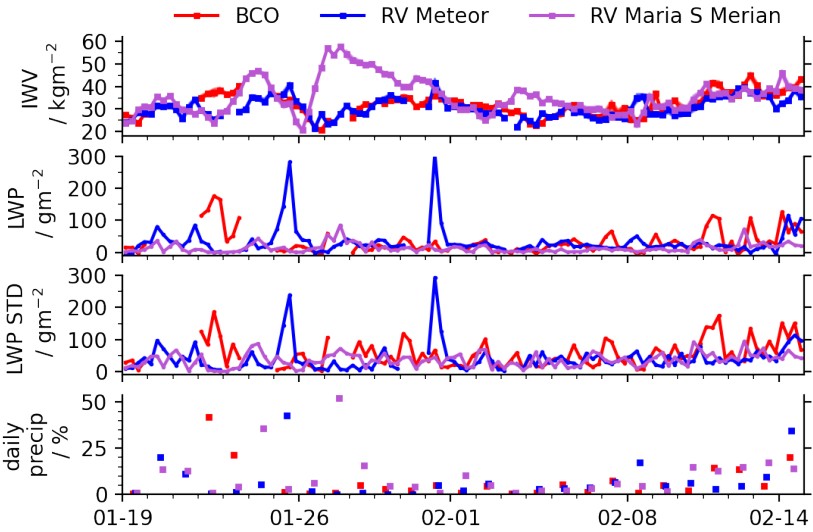

**Figure 3.** Timeline of 6-hourly (a) IWV, (b) LWP, (c) LWP standard deviation in a 6-hour window, (d) daily precipitation fraction, recorded at BCO (red), aboard the Meteor (blue) and Merian (purple) based on the Level-4 data set.

## 5 Integrated Water Vapor

This Section presents the Integrated Water Vapor (IWV) conditions as measured by the different instruments at the different sites and uses independent soundings, Global Navigation Satellite System (GNSS) and ERA5 estimates to evaluate the MWR retrievals.

The IWV conditions measured at each site by each instrument are illustrated in Fig 4 and corresponding distribution parameters are summarized in Table 4. At BCO, a mean IWV of $31.8\,\mathrm{kgm^{-2}}$ was measured in the core period with a standard

deviation of $5.0\,\mathrm{kgm^{-2}}$. The conditions measured aboard the Meteor agree within the associated uncertainty with a mean IWV

**Table 4.** Characteristics of IWV conditions measured by each instrument at each site, including number of valid non-precipitating measurements, mean IWV, median IWV, standard deviation (STD) and skewness of IWV probability distribution. Note that single-channel LIMRAD and MSMRAD IWV is retrieved for clear-sky conditions only.

| site | N | mean IWV | median IWV | IWV STD | skewness |
|---|---|---|---|---|---|
| | - | $\mathrm{kgm}^{-2}$ | $\mathrm{kgm}^{-2}$ | $\mathrm{kgm}^{-2}$ | - |
| BCOHAT | 411643 | 31.8 | 31.8 | 5.0 | 0.3 |
| LIMHAT | 629753 | 30.3 | 29.7 | 4.5 | 0.4 |
| LIMRAD | 396974 | 30.2 | 30.1 | 3.5 | 0.1 |
| MSMRAD | 448666 | 33.3 | 32.3 | 6.3 | 0.6 |

of $30.3\,\mathrm{kgm}^{-2}$, but show slightly less variability (standard deviation of $4.5\,\mathrm{kgm}^{-2}$). The mean conditions aboard the Meteor measured by the LIMHAT and LIMRAD agree, while the LIMRAD IWV distribution is slightly narrower and less skewed due to the fact that the retrieval is only applied in clear-sky conditions. As the Merian was additionally sampling further South over warmer waters with deeper convection, IWV conditions were moister with a mean IWV of $33.3\,\mathrm{kgm}^{-2}$. High IWV conditions

of more than $50\,\mathrm{kgm}^{-2}$, untypical for winter Trade conditions, were observed close to Brazil from January 27 until 29, 2020, associated with a deep convective system. The skewness of all distributions indicates that the 2-month IWV conditions follow a lognormal distribution rather than a normal distribution which is also confirmed visually in Fig 4.

These results align with the results by Foster et al. (2006) who find lognormal distributions in IWV at many locations worldwide, in particular in the (sub-)tropics. EUREC[4]A was slightly moister compared to the dry season conditions observed

during Narval-1 with a mean IWV of $28\,\mathrm{kgm}^{-2}$ (Jacob et al., 2019). Airborne mean IWV of $33.2\,\mathrm{kgm}^{-2}$ measured by the HAMP radiometers aboard HALO (Jacob et al., 2019) is higher than the ground-based estimates from Meteor which sampled a similar area which we relate to the different retrievals used.

We evaluate retrieved IWV by means of the root-mean-square-difference (RMSD), Pearson correlation coefficient, and bias (independent measurement - MWR) with independent IWV measurements derived from radiosoundings (Stephan et al.,

2021) and GNSS (Bock et al., 2021; Bosser et al., 2021), and compare to ERA5 re-analysis data (Fig 5 and Table 5). MWR and radiosoundings are compared in a 10-minute window around each 4-hourly sounding launch to minimize radiosounding drifting effects when comparing to the zenith column. GNSS and MWR measurements are averaged and compared in 15 minute time windows matching GNSS realistic temporal resolution. For the ERA5 intercomparison, MWR measurements are resampled to the full hour and compared to the closest ERA5 field.

Retrieved IWV is closely correlated with sounding IWV at all sites with correlation coefficients higher than 0.9. The RMSD for the HATPRO measurements at BCO is $1.1\,\mathrm{kgm}^{-2}$, which is similar to the MWR-sounding RMSD that Steinke et al. (2015) find. The MWR measurements are on average drier than the radiosoundings' IWV as seen by the positive bias of $1.7\,\mathrm{kgm}^{-2}$. A similar bias of $1.6\,\mathrm{kgm}^{-2}$ is found in the LIMHAT - sounding comparison, although the RMSD is smaller than at BCO

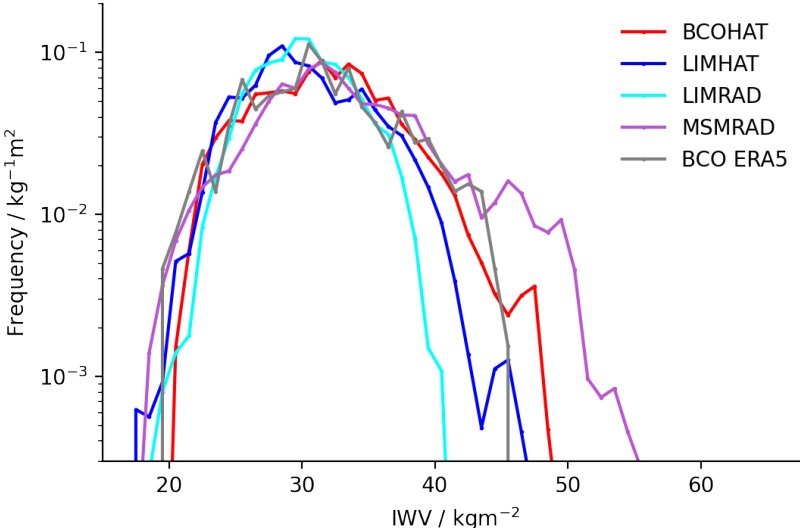

**Figure 4.** Frequency of Occurrence of IWV retrieved from HATPRO measurements at BCO (BCOHAT, red) and aboard Meteor (LIMHAT, blue), as well as from single-channel $T_B$ aboard Meteor (LIMRAD, cyan) and Merian (MSMRAD, purple). The distribution of ERA5 values at BCO (gray) is added for comparison. Displayed frequencies are cut if calculated from less than 30 measurement points. Note that LIMRAD and MSMRAD IWV is only retrieved in clear-sky conditions.

($0.7\,\mathrm{kg m^{-2}}$). The dry bias between MWR measurements and radiosoundings at both BCO and Meteor could be related to the

fact, that the statistical retrieval is trained on radiosoundings launched from Grantley International Airport. Bock et al. (2021) find that the airport radiosoundings exhibit a dry bias of $2.9\,\mathrm{kg m^{-2}}$ compared to the Vaisala MW41 radiosoundings used at BCO during EUREC[4]A (Stephan et al., 2021). Aboard the Meteor, the LIMHAT IWV data set can additionally be used to evaluate the dropsondes launched from HALO's circles (George et al., 2021) which were corrected for a dry bias compared to the radiosounding data set.

The MWR-sounding bias of clear-sky IWV retrieved from LIMRAD is reduced by $70\,\%$ compared to the respective HATPRO derived IWV. The RMSD of LIMRAD - radiosoundings (1.3) is slightly smaller than at BCO, while, the Merian measurements' RMSD is higher than expected ($3.6\,\mathrm{kg m^{-2}}$). This increase in RMSD might be related to the lower number of radiosoundings used for training and evaluation which could also explain the switch of bias sign to negative values. Single- and multi-channel clear-sky IWV retrievals can be directly intercompared using simultaneous LIMRAD and LIMHAT measurements aboard the

Meteor. All core period measurements agree with an RMSD of $1.2\,\mathrm{kg m^{-2}}$, affected by a bias of $1\,\mathrm{kg m^{-2}}$ with LIMRAD being moister than LIMHAT.

At BCO, IWV obtained from GNSS and BCOHAT exhibit a RMSD of $1.4\,\mathrm{kg m^{-2}}$. As opposed to Bock et al. (2021), we do not find a bias between the measurements which could be attributed to different quality filtering mechanisms used in this analysis. Aboard the Meteor, the LIMHAT - GNSS RMSD is similar ($1.4\,\mathrm{kg m^{-2}}$) but affected by a negative bias of $-1.1\,\mathrm{kg m^{-2}}$

**Table 5.** Evaluation of MWR retrieved IWV from BCOHAT, LIMHAT, LIMRAD and MSMRAD relative to independent IWV measurements of radiosoundings, GNSS and closest ERA5 field through RMSD, bias and correlation coefficient. A positive bias refers to drier MWR conditions than measured by the respective independent IWV measurement. Note that LIMRAD and MSMRAD evaluation is performed in clear-sky conditions only.

| | | sounding | GNSS | ERA5 |
|---|---|---|---|---|
| | N | 125 | 2013 | 514 |
| BCO | RMSD | 1.1 | 1.4 | 2.2 |
| BCOHAT | bias | 1.7 | -0.1 | -1.0 |
| | corr | 0.97 | 0.96 | 0.90 |
| | N | 164 | 2377 | 427 |
| Meteor | RMSD | 0.7 | 1.5 | 2.3 |
| LIMHAT | bias | 1.6 | -1.1 | -0.1 |
| | corr | 0.99 | 0.95 | 0.86 |
| | N | 120 | 1779 | 394 |
| Meteor | RMSD | 1.3 | 1.8 | 2.4 |
| LIMRAD | bias | 0.5 | -1.9 | -0.7 |
| | corr | 0.97 | 0.90 | 0.79 |
| | N | 82 | 1632 | 392 |
| Merian | RMSD | 3.6 | 6.5 | 2.9 |
| MSMRAD | bias | -0.5 | -3.8 | -1.3 |
| | corr | 0.91 | 0.72 | 0.93 |

with GNSS measurements drier than the MWR measurements. Bosser et al. (2021) report that the GNSS measurements aboard the Merian were of poor quality which explains the large RMSD and bias when comparing to MSMRAD IWV.

The two periods of ship collocation (see Sec. 2.2) allows a direct comparison of clear-sky IWV derived from LIMRAD, LIMHAT and MSMRAD. Comparing the radars from both ships, LIMRAD and MSMRAD are associated with an RMSD of $1.1\,\mathrm{kg\,m^{-2}}$, a correlation coefficient of 0.88, and a bias of -0.3 (LIMRAD moister than MSMRAD). Given this good agreement, MSMRAD IWV seems more accurate than the GNSS measurements, and closes the measurement gap of highly temporally resolved IWV measurements aboard the Merian.

As MWR measurements were not assimilated into re-analysis, a comparison to re-analysis ERA5 fields closest in time and space can provide further retrieval evaluation. Retrieved IWV and ERA5 RMSD at BCO and Meteor agree to within $2.5\,\mathrm{kg\,m^{-2}}$ with slightly higher agreement aboard the Merian ($2.9\,\mathrm{kg\,m^{-2}}$). While ERA5's IWV is unbiased compared to LIMHAT IWV aboard the Meteor, it is dry biased by $-1.0\,\mathrm{kg\,m^{-2}}$ and $-1.3\,\mathrm{kg\,m^{-2}}$ at BCO and aboard Merian, respectively.

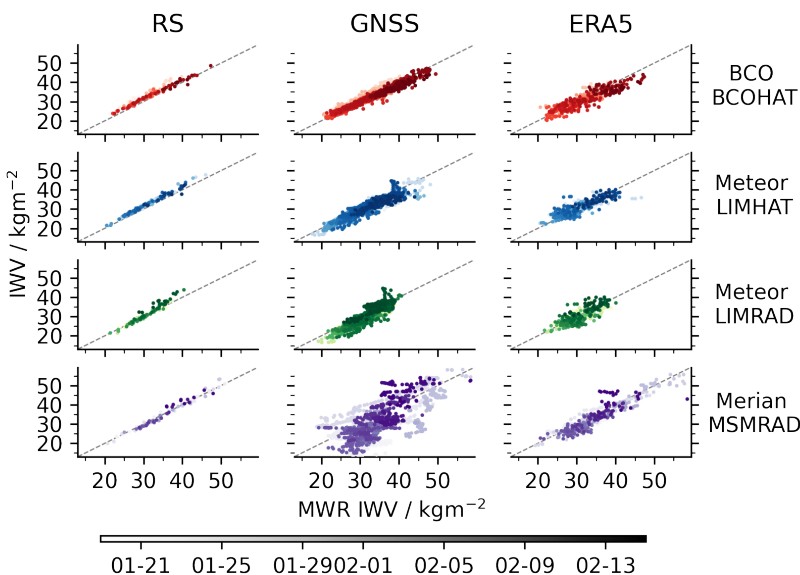

**Figure 5.** Pair-wise IWV evaluation of MWR retrieved IWV (x-axis) for four different instruments (rows) to independent measurements (y-axis) of radiosoundings (first column), GNSS (second column), ERA5 (third column), color-coded by time from January 19, 2020 (light) until February 14, 2020 (dark). Note that IWV from LIMRAD and MSMRAD is only available in clear-sky conditions.

## 6 Liquid Water Path

This Section describes the Liquid Water Path (LWP) conditions retrieved from the different instruments in non-precipitating conditions. Separating conditions in clear-sky and cloudy sky requires the cloud mask introduced in Sec 4.1. The resulting liquid cloudy LWP conditions are analyzed in Sec 6.1. Clear-sky identified scenes serve as base to characterize the clear-sky LWP noise, contributing to the overall LWP uncertainty and detection limit analysis presented in Sec 6.2. Inter-platform retrieval comparison is performed for two limited time periods in which the Meteor and Merian were measuring in close proximity (Sec. 6.3).

### 6.1 Cloudy LWP

Liquid cloud LWP is analyzed by applying the joint cloud mask (Sec. 4.1) to retrieved LWP. Fig 6 illustrates retrieved LWP distributions observed by BCOHAT, LIMHAT, LIMRAD and MSMRAD in confident liquid cloudy scenes. Corresponding distribution parameters are summarized in Table 6. Mean LWP conditions in confident liquid cloudy conditions at BCO, aboard the Meteor and Merian were 63.1, 62.5, 52.4 and 46.8 $gm^{-2}$. The mean conditions at BCO and Meteor align well with the mean airborne LWP of 63 $gm^{-2}$ observed during NARVAL-1 in similarly dry winter trade conditions across the same region (Jacob et al., 2019; Schnitt et al., 2017). BCOHAT and LIMHAT retrieved mean LWP of 63.1 $gm^{-2}$ and 62.5 $gm^{-2}$ agree well within their associated LWP uncertainties (see Sec 6.2). Even though similar mean LWP conditions were observed, more detailed

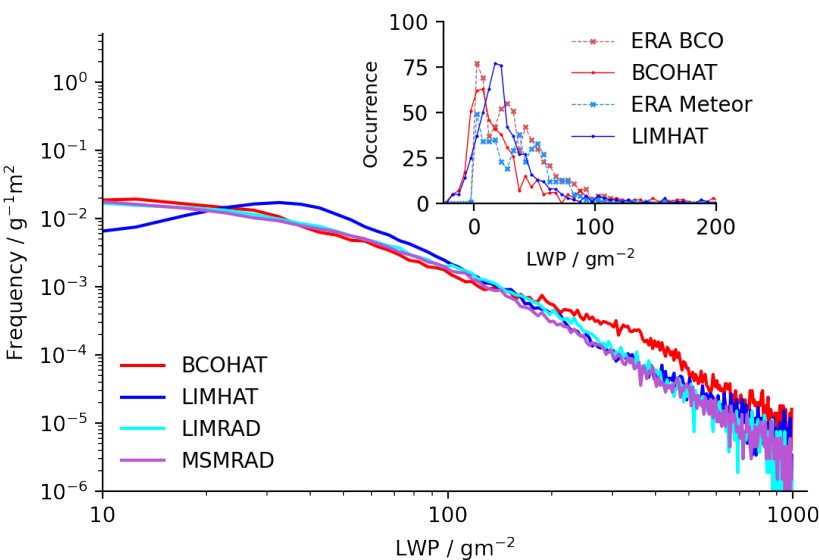

**Figure 6.** Distribution of LWP occurrence in confident cloudy, non-precipitating scenes at BCO (red), aboard the Meteor retrieved from LIMHAT (blue) and LIMRAD (cyan) measurements, as well as aboard the Merian with MSMRAD (purple). The inlet displays the distribution of LWP re-sampled to the full hour from BCOHAT (red, solid) and LIMHAT (blue, solid), and the corresponding hourly-resolved ERA5 total column liquid water (dashed).

**Table 6.** Characteristics of non-precipitating LWP distribution, including mean, median, standard deviation, 10th and 90th percentile, skewness, as retrieved from BCOHAT, LIMHAT, LIMRAD and MSMRAD in confident liquid cloudy (confident and probably liquid cloudy) identified scenes.

| site | cloud cover | mean | median | standard dev | 10th | 90th | skewness |
|------|-------------|------|--------|--------------|------|------|----------|
| | % | $gm^{-2}$ | $gm^{-2}$ | $gm^{-2}$ | $gm^{-2}$ | $gm^{-2}$ | - |
| BCOHAT | 33.2 (46.5) | 63.1 (62.6) | 27.5 (25.9) | 104.3 (106.4) | 3.8 (2.2) | 163.9 (167.8) | 3.8 (3.8) |
| LIMHAT | 19.3 (22.5) | 62.5 (63.4) | 41.6 (41.1) | 78.0 (79.5) | 15.3 (13.2) | 121.8 (131.3) | 5.0 (4.6) |
| LIMRAD | 21.5 (23.0) | 52.4 (50.0) | 28.5 (26.1) | 77.4 (76.3) | 2.6 (1.3) | 125.2 (122.6) | 4.3 (4.3) |
| MSMRAD | 21.0 | 46.8 | 24.4 | 74.6 | 0.9 | 110.5 | 4.7 |

trajectory analyses are necessary to investigate the effect of ocean surface and island impact on the cloud evolution between Meteor and BCO. Median and mean LWP differ as the mean LWP is influenced by single events of high LWP, e.g. through un-flagged precipitation or sea-spray, while the median is driven by the large amount of small LWP below the instruments' detection limit.

90 % of observed confident liquid cloudy columns were associated with a LWP of around 160, 120 and 110 gm$^{-2}$ at BCO, and aboard the Meteor and Merian, respectively. The comparatively higher LWP 90th percentile and standard deviation at BCO are most probably related to wet radome conditions as the blower unit of BCOHAT was broken throughout some of the core period (as opposed to the other instruments). We also suspect that the sea-spray altered, aged radome was less hygroscopic compared to the newer LIMHAT radome, leading to additional moisture on the radome and longer drying times. An additional

island impact triggering deeper convection in prevailing non-trade-wind conditions is in ongoing analysis. The close to zero 10th percentile reflects the fact that the statistical regression covers negative values to avoid biasing the overall distribution (see Sec 3.1), and indicates that the cloud mask did not perform perfectly well in all conditions. Likely, the wider field of view of the MWR compared to the active remote sensing instruments used for the cloud mask led to mis-identification of scenes.

Expanding the analysis to include *probably liquid cloudy* conditions slightly reduces mean and median LWP, as well as all

other parameters of the distribution given in Table 6, likely due to the fact that more mis-flagged clear-sky conditions impact the LWP distribution. This shift in distribution parameters illustrates the sensitivity of the derived LWP properties to the cloud mask performance.

At BCO and Meteor, retrieved LWP is compared to the ERA5 estimates by resampling BCOHAT and LIMHAT's LWP, respectively, to every hour (see inlet in Fig 6). Instrument-derived mean LWP of 33.2 and 39.5 gm$^{-2}$ agree well with ERA5

mean LWP of 34.5 gm$^{-2}$ and 36.0 at BCO and Meteor, respectively. The measured LWP variability, here quantified as standard deviation, of 58.0 and 84.7 gm$^{-2}$ is higher than the ERA5 variability of 27.9 and 26.4 gm$^{-2}$, respectively, which we attribute to the horizontal resolution of ERA and the small cloud sizes.

## 6.2 LWP Uncertainty and Detection Limit

Characterizing the uncertainty of the retrieved LWP by independent measurements is not straight forward as LWP retrieved

from measurements by visible or infrared remote sensing techniques is not sensitive to the same column as the microwave measurements (e.g. Turner et al., 2007b). Therefore, a clear-sky LWP noise can be derived by analyzing retrieved LWP in independently classified clear-sky cases as a generally accepted strategy (Jacob et al., 2019; van Meijgaard and Crewell, 2005). Retrieval offsets to zero are due to the statistical nature of the retrieval approach, due to calibration artefacts, and radiometric noise. The lowest detectable LWP is then calculated from the clear-sky LWP noise for different water vapor conditions. Cloudy-

sky LWP uncertainty can be estimated as a function of LWP by calculating a root-mean-square difference (RMSD) of true versus retrieved LWP. True LWP here refers to the LWP used to forward-model $T_B$ in the radiative transfer calculations (see Sec 3), while retrieved LWP is the result of applying the respective retrieval equation to the same $T_B$.

The retrieved clear-sky LWP distribution at BCO is illustrated in Fig 7a, and Table 7 summarizes the distribution character-istics for all sites. 49.0, 59.0, 61.1 and 75.2 % of all valid LWP BCOHAT, LIMHAT, LIMRAD and MSMRAD measurements,

respectively, are identified as clear-sky. Note that the fractions disagree for LIMRAD and LIMHAT aboard the Meteor due to different observational gaps in the measurements. Applying a Gaussian fit to the distribution yields to a mean and standard de-viation, which is interpreted as clear-sky LWP bias and clear-sky LWP noise, respectively. The Gaussian fit widths of 9.9 gm$^{-2}$ and 12.0 gm$^{-2}$ for BCOHAT and LIMHAT, respectively, quantify the clear-sky LWP noise, and match clear-sky noises pre-

**Table 7.** Parameters of clear-sky LWP distribution at all sites, including clear-sky fraction of all valid LWP measurements, median, mean, standard deviation, 10th and 90th percentile. Additionally, mean and standard deviation of a Gaussian fit are given.

| site | clear-sky % | mean $\mathrm{gm}^{-2}$ | median $\mathrm{gm}^{-2}$ | standard dev $\mathrm{gm}^{-2}$ | 10th $\mathrm{gm}^{-2}$ | 90th $\mathrm{gm}^{-2}$ | fit mean $\mathrm{gm}^{-2}$ | fit standard dev $\mathrm{gm}^{-2}$ |
|---|---|---|---|---|---|---|---|---|
| BCOHAT | 49.0 | 3.9 | 2.1 | 20.4 | -9.0 | 16.2 | 2.7 | 9.9 |
| LIMHAT | 59.0 | 11.5 | 12.6 | 12.1 | -4.4 | 25.3 | 11.5 | 12.0 |
| LIMRAD | 61.1 | -0.4 | 0.0 | 4.0 | -3.1 | 1.2 | -0.4 | 3.4 |
| MSMRAD | 75.2 | 0.6 | 0.0 | 8.7 | -4.4 | 5.2 | 0.0 | 4.5 |

**Table 8.** Characteristics of confident liquid cloudy Level 3 LWP distribution considering each instrument's detection limit. Fraction (relative to all valid confident liquid cloudy measurements) and mean LWP are calculated for the following LWP bins: LWP below detection threshold, LWP between detection threshold and $30\,\mathrm{gm}^{-2}$, LWP between 30 and $100\,\mathrm{gm}^{-2}$, and LWP above $100\,\mathrm{gm}^{-2}$.

| | detection limit $\mathrm{gm}^{-2}$ | LWP < detect fraction % | LWP < detect mean $\mathrm{gm}^{-2}$ | detect < LWP < 30 fraction % | detect < LWP < 30 mean $\mathrm{gm}^{-2}$ | 30 < LWP < 100 fraction % | 30 < LWP < 100 mean $\mathrm{gm}^{-2}$ | LWP > 100 fraction % | LWP > 100 mean $\mathrm{gm}^{-2}$ |
|---|---|---|---|---|---|---|---|---|---|
| BCOHAT | 9.9 | 20.8 | 2.4 | 32.5 | 19.1 | 30.4 | 54.0 | 16.3 | 244.9 |
| LIMHAT | 12.0 | 7.3 | 4.7 | 23.5 | 22.2 | 55.3 | 52.9 | 13.9 | 199.2 |
| LIMRAD | 3.4 | 11.5 | -1.0 | 40.1 | 15.4 | 34.1 | 55.3 | 14.2 | 193.5 |
| MSMRAD | 4.5 | 18.2 | -0.8 | 37.6 | 15.5 | 32.5 | 55.5 | 11.7 | 196.5 |

viously identified for retrievals based on the similar channels (Jacob et al., 2019; Schnitt et al., 2017). The single-channel clear-sky LWP noises are smaller (3.4 and $4.5\,\mathrm{gm}^{-2}$, respectively), as IWV is fixed due to retrieving from the $T_{\mathrm{B}}$ difference of cloudy and clear-sky; and as water vapor absorption is stronger at $89\,\mathrm{GHz}$ compared to the lower frequencies used in BCOHAT and LIMHAT. The lowest detectable LWP depends on the vertical water vapor distribution which, in cloudy conditions, is not available at any of the sites. Therefore, we estimate the smallest detectable LWP as the clear-sky LWP noise which, in turn, depends on the performance of the independent cloud masking algorithm.

Quantifying the detection limit allows to analyze which clouds are missed by the different radiometers. 79.2, 92.8, 88.5 and 81.8 % of all confident liquid cloudy flagged measurements contain LWP above the respective detection limits of BCOHAT, LIMHAT, LIMRAD and MSMRAD (see Table 8). The remaining undetected LWP compared to the ceilometer-radar cloud mask is most likely associated to optically thin clouds with low water contents (e.g. Mieslinger et al., 2022) and to cloud mask performance. This reduction in cloud cover when derived from passive microwave sensors is also observed by the airborne cloud masks (Konow et al., 2021). One third of detected LWP is seen between detection limit and $30\,\mathrm{gm}^{-2}$, as well as between

30 and 100 gm$^{-2}$, averaging to mean LWP conditions of 19 to 22 gm$^{-2}$ and around 55 gm$^{-2}$, respectively. Only 11 to 16 % of detected LWP at BCO and aboard the ships, respectively, are associated with thicker clouds of higher than 100 gm$^{-2}$.

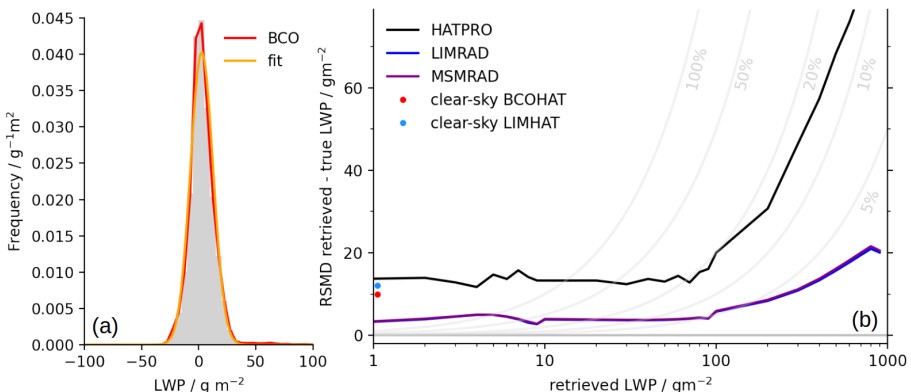

**Figure 7.** (a) Distribution of occurrence of BCOHAT LWP in clear-sky identified scenes (red) and respective Gaussian fit (orange), and (b) RMSD of retrieved vs true LWP for HATPRO (black) and single-channel retrieval (purple, blue), binned to retrieved LWP. Respective clear-sky Gaussian standard deviations are given for BCOHAT (red) and LIMHAT (blue).

Cloudy LWP uncertainty varies as function of retrieved LWP as illustrated in Fig 7b) binned to logarithmic bins of LWP. The mean RMSD for HATPRO derived LWPs below 20 gm$^{-2}$ varies below 5 gm$^{-2}$, corresponding to a relative RMSD between
410 75 and 50 %. For LWP between 20 and 100 gm$^{-2}$, the RMSD moderately reduces from 50 to 15 % of retrieved LWP (15.8 at 50 gm$^{-2}$). Above LWP of 100 gm$^{-2}$, the relative uncertainty is better than 15 % (e.g. 29.9 gm$^{-2}$ at LWP of 200 gm$^{-2}$). Higher LWP values are in reality often affected by precipitation and, thus, not sensed by ground-/ship-based MWR measurements. Jacob et al. (2019) find on average higher RMSDs which we relate to the additional uncertainty given by the background emission characterization for airborne LWP retrieval.
The single-channel retrieval, different in retrieval design and training data set compared to the multi-channel retrieval, is characterized by lower uncertainties and detection limit. Higher liquid water emissions in the 89 GHz channel compared to the 31.4 GHz channel used in the multi-frequency HATPRO retrieval leads to a higher sensitivity of the retrieval to smaller clouds with less liquid. This retrieval, however, strongly depends on the knowledge of IWV conditions and accurate clear-sky flagging. Relative LWP uncertainty for LWP between 10 and 100 gm$^{-2}$ increases by few percentage points if closest clear-sky
and cloudy IWV differ by 1 to 2 kgm$^{-2}$.

### 6.3 Single-channel retrieval intercomparison

The availability of both multi- and single-channel retrieval aboard the Meteor allows a direct comparison of the two different retrieval approaches. A direct intercomparison in confident liquid cloudy conditions reveals a LWP RMSD of 31.3 gm$^{-2}$, a bias of -5.9 gm$^{-2}$ (LIMHAT LWP higher than LIMRAD), and a high correlation of 0.92 (not shown). As LWP varies strongly

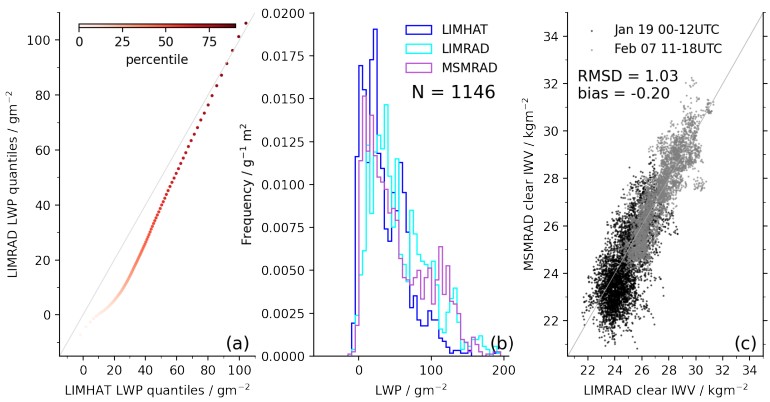

**Figure 8.** (a) Percentiles of LIMHAT and LIMRAD retrieved liquid cloudy LWP distributions during EUREC[4]A core period. Intercomparison of LIMRAD and MSMRAD retrieved (b) liquid cloudy LWP and (c) clear-sky IWV when Meteor and Merian steamed along the same trajectory (January 19, 2020, 00-12UTC), and measured at the same location (February 07, 2020, 11-18UTC).

in time and space and sensors's fields of view are different, comparing the liquid cloudy LWP distributions through percentiles is a preferable method. The percentiles of the liquid cloudy LWP distribution of LIMRAD and LIMHAT, illustrated in Fig 8, show that LIMRAD-retrieved LWP is skewed to lower values compared to LIMHAT's LWP. The different clear-sky correction approaches in the two retrievals constitute themselves in the fact that LIMRAD LWP approaches zero when LIMHAT LWP ranges between 5 and $17 \, \mathrm{gm}^{-2}$. Above this range, the negative bias towards LIMRAD, showing less LWP than LIMHAT, moderately decreases towards higher LWP values.

Inter-platform evaluation of single-channel retrieved LWP and clear-sky IWV is performed for the two periods of ship collocation (see Sec. 2.2). Ships's visiting times at BCO could not be used as BCOHAT was not operational at those times. LWP obtained from LIMHAT, LIMRAD and MSMRAD are intercompared in a statistical way rather than directly as clouds might overpass with an unknown time shift. Both LIMRAD and MSMRAD exhibit larger LWPs (median of 50.4 and $42.6 \, \mathrm{gm}^{-2}$, respectively) than LIMHAT ($28.4 \, \mathrm{gm}^{-2}$), confirming the percentile based comparison of LIMRAD and LIMHAT. Cloudy profiles of above $100 \, \mathrm{gm}^{-2}$ were mostly seen by MSMRAD, which, however, might be related to single events that did not overpass the Meteor given the small sample size. Additionally, both radars were operated with different chirp table settings, leading to different sensitivity to boundary layer clouds which, in turn, might affect the performance of the cloud mask. Given the uncertainties of each LWP product identified in the previous Section and the uncertainty related to the applied cloud mask, the distributions match well and are suitable for site intercomparison. Clear-sky IWV, less variable in space and time, is compared point to point, and exhibits a RMSD of 1.0 and a bias of $-0.2 \, \mathrm{kgm}^{-2}$ (MSMRAD slightly drier). Both single-channel retrievals agree within the expected uncertainties.

Assuming that BCO and Meteor were exposed to similar conditions and given the fact, that multi-channel derived LWP is generally more reliable, we conclude that the LIMHAT measurements should be used as 'truth' for the Meteor site compared

to the single-channel LIMRAD LWP. While LIMRAD single-channel LWP is biased by -5.9 g m$^{-2}$ compared to the multi-channel LWP estimates, presumably due to a higher sensitivity towards smaller clouds, this bias cannot be directly translated to MSMRAD LWP due to absolute calibration differences of the two cloud radars. Clear-sky $T_B$ are affected by a RMSD of 2.6 K, a bias of 5.6 K (Merian warmer), but correlation of 0.66 is low due to temporal spatial mismatch. Given this $T_B$ bias and assuming all other instrument characteristics being the same between LIMRAD and MSMRAD, Merian single-channel LWPs might in reality be lower. An extended analysis can help to quantify this bias, e.g. by comparing similar looking clouds as seen in the active cloud radar part.

## 7 Thermodynamical profiles

The multi-channel measurements by the HATPRO instruments are used to retrieve temperature (see Sec 7.1) and absolute humidity (see Sec 7.2) profiles at BCO and aboard the Meteor, respectively. Temperature profiles are obtained from zenith measurements and when elevation scans were performed (see Sec 3.1), while absolute humidity profiles are only available in zenith mode. Profiles are obtained on 43 height levels with vertical resolution decreasing from 50-100 m in the moist layer to 200-500 m above the trade inversion. We use the EUREC$^4$A sounding Level-2 data set (Stephan et al., 2021) to evaluate the MWR retrieved profiles, assuming that the radiosoundings represent the best estimate of the true atmospheric conditions. To compare radiosoundings and MWR, we interpolate the radiosoundings to the MWR height grid and average MWR measurements 5 minutes around each sounding launch as conditions in the Tropics change on longer timescales. 182 and 219 radiosoundings are used for BCO and Meteor, respectively. We then calculate RMSD and bias for each MWR height level. Positive biases here indicate an overestimation of MWR compared to the sounding value.

### 7.1 Temperature

The obtained temperature RMSD and bias are illustrated in Fig 9. At both sites, zenith mode RMSD increases throughout the moist layer from less than 0.5 K below Lifting Condensation Level (LCL) to 1.5 K at the trade inversion around 2 km. As zenith HATPRO measurements generally contain two degrees of freedom (independent pieces of information) for retrieving the temperature profile (Löhnert et al., 2009), the retrieval information content is too low to resolve the trade temperature inversion. Rather, the MWR profiles smooth the inversion, resulting in on average warmer MWR conditions at the base of the inversion, and colder conditions at inversion top, similar to conditions found in the Arctic (Walbröl et al., 2022). Temperature information content is highest below 4 km (Löhnert and Maier, 2012), which makes the MWR insensitive to the conditions in the middle troposphere as seen by further increasing RMSD.

Elevation scans have been shown to improve the derived temperature profile in the lowest kilometer of the boundary layer (Crewell and Löhnert, 2007; Walbröl et al., 2022). As illustrated in Fig 9, however, the BCOHAT and LIMHAT scans increase RMSD and bias in the layers below 1 km. We suspect that the GAIA sounding data set used for training is impacted by the island surface, leading to warmer temperatures in the moist layer compared to the zenith column at BCO or over the ocean. Typically, when trade winds prevail, radiosoundings launched at GAIA or BCO drift westwards over the island when ascending

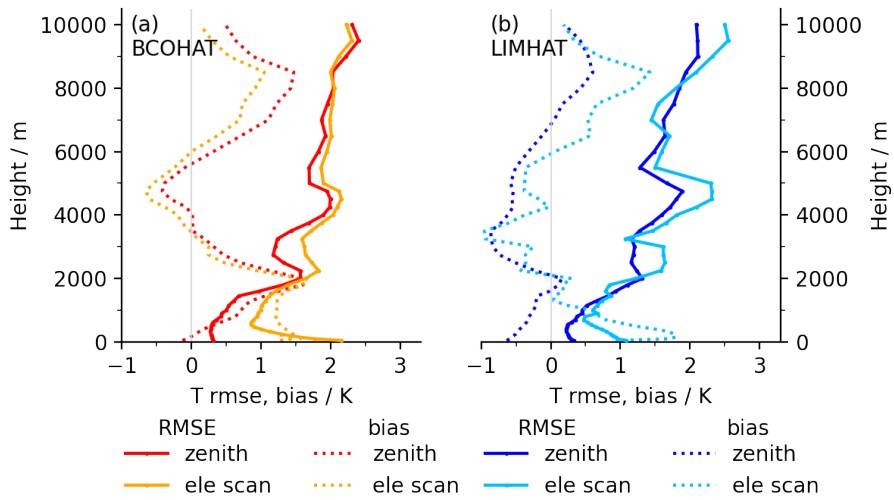

**Figure 9.** RMSD (solid) and bias (dashed) of (a) BCOHAT and (b) LIMHAT temperature profiles from zenith (blue; red) and elevation scan (orange; cyan) operation compared to simultaneous sounding profiles.

through the sub-cloud layer. Paired with small seasonal temperature variations in the Tropics and, thus, little variability in the temperature training data set, this systematic training error translates into warm temperature biases of the retrieved temperature profiles compared to the launched radiosoundings. A physical-iterative retrieval approach such as Optimal Estimation (Rodgers, 2000; Maahn et al., 2020) would help to constrain the covariances of the prior temperature profile data set. Elevation scans aboard the Meteor, and therein in particular the low elevation angle measurements, are additionally affected by ship motion as LIMHAT was not stabilized. The functionality of the HATPRO-attached weather station might additionally impact the quality of the temperature retrieval.

## 7.2 Absolute Humidity

Comparing radiosoundings and MWR yields to the RMSD and bias illustrated in Fig 10(a). At both sites, the RMSD from ground to LCL is $1.3\,\mathrm{g\,m}^{-3}$, and increases to $2.5\,\mathrm{g\,m}^{-3}$ in the area of the hydrolapse associated with the trade inversion. The tendencies of the bias can be further understood when analyzing the mean profiles as illustrated in Fig 10(b). From ground towards hydrolapse, MWR underestimates the humidity, resulting in a negative bias of $-1\,\mathrm{g\,m}^{-3}$. Throughout the hydrolapse, MWR and sounding profiles converge, which is due to the smoothing of the MWR profile. Depending on the strength of the hydrolapse, MWR overestimates the humidity in the dry layer balancing the overall profile to match overall IWV conditions. Above the hydrolapse in the free troposphere, dry conditions prevail, and MWR is not sensitive to elevated moist layers. While the MWR covers the variability of moist layer water vapor well as seen by similar standard deviations of sounding and

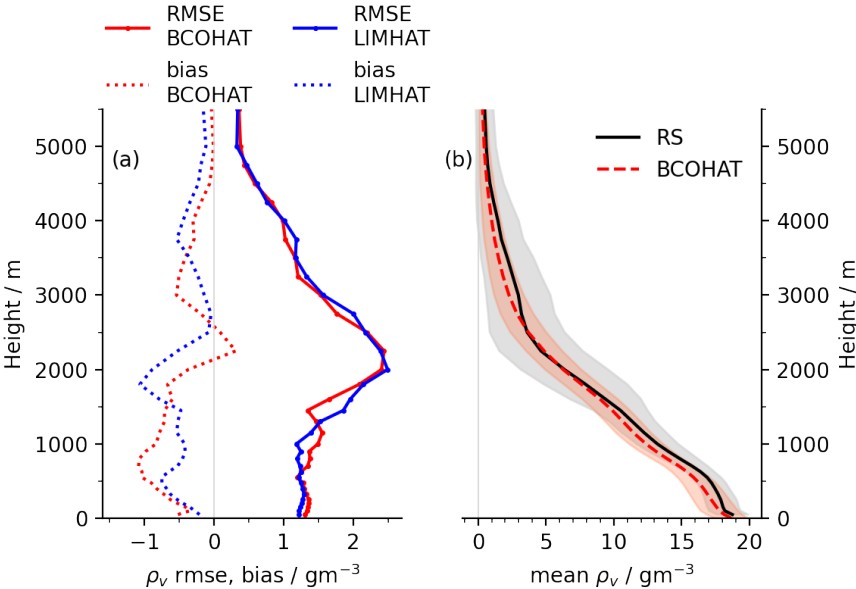

**Figure 10.** (a) BCOHAT (red) and LIMHAT (blue) RMSD (solid) and bias (dashed) of retrieved absolute humidity $\rho_v$ profiles compared to simultaneous sounding profiles, and (b) mean $\rho_v$ profiles of radiosoundings (black) and BCOHAT (red), shaded by their respective standard deviation.

MWR profile, it does not resolve the variability in the hydrolapse or free troposphere. The overall negative bias in the absolute humidity profile translates into a dry bias in the IWV estimate (compared to the radiosoundings) which confirms the findings in Sec 5.

## 8 Conclusions

This study presents the ground- and ship-based passive MWR measurements performed during the EUREC[4]A field study. Between January 19 and February 14, 2020, continuous measurements of IWV, LWP, and coarse profiles of temperature and absolute humidity were obtained in the vicinity of Barbados at 3 second resolution. 14-channel MWR measurements were performed at Barbados Cloud Observatory and aboard the Meteor with a HATPRO microwave radiometer, while single-channel measurements were performed at 89 GHz aboard the Meteor and the Merian complementing W-band cloud radar measurements.

The here presented data set contributes key measurements to study the coupling of clouds to cirulcation and their environment, the overall goal of the EUREC[4]A field study (Bony et al., 2017; Stevens et al., 2021). The data set enables a continuous quantification of clouds' LWP in their immediate moisture environment, enables the characterization along spatial scales across the trade-driven Tropical Atlantic, and complements the airborne LWP measurements performed aboard HALO and the SAFIRE ATR42.

Similar mean IWV conditions of 31.8 and 30.3 kgm$^{-2}$ at BCO and aboard the Meteor, respectively, support the hypothesis that similar air masses were observed, evolving from Meteor towards BCO along the trade-wind driven region. The Merian sampled moister conditions on its track southward, leading to mean IWV conditions of 33.3 kgm$^{-2}$. The multi-channel retrieved IWV at BCO is affected by a RMSD of 1.1, 1.4 and 2.2 kgm$^{-2}$ compared to radiosoundings, GNSS and ERA5 estimates, matching uncertainties identified in mid-latitudes (Steinke et al., 2015).

A precipitation and cloud mask are included in the data set, as derived from adjacent weather station and simultaneous cloud radar and ceilometer measurements. Cloudy scenes are additionally flagged for liquid cloud occurrence based on the radar observations. We find that 9.1, 10.7 and 14.6 % of all valid measurements contain ground-reaching precipitation at BCO, Meteor, Merian, respectively. Confident liquid cloudy scenes prevail in 33.5, 19.3 and 21.0 % of available profiles, respectively, matching cloud cover estimates in Nuijens et al. (2014). Confident liquid cloudy LWP distributions reveal a mean LWP of 63.1, 62.5 and 46.8 gm$^{-2}$ at BCO, Meteor and Merian, respectively, which align with findings in Jacob et al. (2019). 90 % of all confident liquid cloudy profiles contained around 160 and 120 gm$^{-2}$ LWP at BCO and aboard the Meteor and Merian, respectively. Derived LWP statistics depend on the performance of the cloud masking algorithm. When including probably cloudy identified scenes in the statistics, mean LWP and percentiles reduce slightly due to beam mismatches and resulting mis-identification of clear scenes. Multi-channel retrieved LWP at BCO and aboard the Meteor is provided with an uncertainty of 30 % at 50 gm$^{-2}$ and better than 15 % above 100 gm$^{-2}$. Single-channel retrieved LWP uncertainty is reduced by 70 % at 50 gm$^{-2}$ but might in reality be higher as the retrieval requires accurate quantification of IWV and clear-sky identification. Clear-sky LWP noise reveals a detection limit of 9.9, 12.0, 3.4 and 4.5 gm$^{-2}$ for BCOHAT, LIMHAT, LIMRAD and MSMRAD. Up to 20 % of confident liquid cloudy tagged profiles are below the LWP detection limit presumably due to undetected optically thin clouds (Mieslinger et al., 2022).

We recommend using the Level 4 data set for non-expert users as quality and precipitation flags were applied to the provided IWV and LWP time series. Data are re-sampled to different temporal resolutions, facilitating model-observation intercomparison experiments. More experienced users will find more details in the Level 3 data set, including a liquid cloud flag and the temperature and humidity retrieval output. Future retrieval approaches could combine HATPRO and 89 GHz channel (Crewell and Löhnert, 2003) to advance the retrieval performance. More specifically, improvements are expected by applying neural network based (e.g. Jacob et al., 2019; Cadeddu et al., 2009) or physical (e.g. Löhnert et al., 2004; Turner et al., 2007a; Maahn et al., 2020) retrieval approaches. The single-channel LWP retrieval can be used to evaluate the approach presented by Billault-Roux and Berne (2021). The spatial dimension of this data set are currently further exploited by characterizing LWP and IWV conditions in different mesoscale organization conditions (e.g. Schulz et al., 2021), and by evaluating microwave and vis/IR satellite LWP products as well as climatologies (Elsaesser et al., 2017). Combining BCO and Meteor measurements can frame Lagrangian trajectory analyses targeting the evolution of air masses along the trade winds. Using this data set to benchmark cloud-resolving simulations will help answering some of the central questions targeted by the EUREC[4]A field study on the interplay of clouds, circulation, convection and climate.

## 9 Code and data availability

The presented data set is available through AERIS (Schnitt et al., 2023a, https://doi.org/10.25326/454#v2.0). Processing and analysis code are available in Schnitt et al. (2023b) (https://doi.org/10.5281/zenodo.8208499).

*Author contributions.*

SaS led the study, developed the Barbados-specific HATPRO retrieval, and prepared data set, manuscript and figures. AF and HKL supported the conceptualization of the study and led the Meteor measurements and post-processing, supported by JR. MM and SC developed and ran the single-channel retrieval. CA led the measurements aboard the Merian. UL provided expertise on HATPRO retrieval and BP contributed the spectral flagging retrieval. FJ maintains the BCO measurements. BS led the EUREC[4]A campaign. All authors contributed to the manuscript.

*Competing interests.*    The authors have no competing interests.

*Acknowledgements.*    The data used in this publication was gathered in the EUREC[4]A field campaign and is made available through Max Planck Institute for Meteorology, University Leipzig, University of Cologne. EUREC[4]A is funded with support of the European Research Council (ERC), the Max Planck Society (MPG), the German Research Foundation (DFG), the German Meteorological Weather Service (DWD) and the German Aerospace Center (DLR). SaS' EUREC[4]A participation was funded by a travel grant from Graduate School of

Geosciences (GSGS), University of Cologne. We thank the BCO team for maintaining the observatory, and the AERIS team, especially Vincent Douet, for their support in publishing the data set. We thank two anonymous reviewers for their helpful comments.

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
