# Peer review of "Ground- and ship-based microwave radiometer measurements during EUREC4A"

_Earth System Science Data, 2023_

## Referee Comment (RC2)

Review of: essd-2023-140
Ground- and ship-based microwave radiometer measurements during EUREC4A
By Schnitt et al.,

This manuscript describes the passive microwave measurements and the LWP and IWV retrievals based on them from multi-wavelength K- and V-band radiometer on the Meteor and at BCO, and single-wavelength radiometer retrievals using radar-determined cloud identification on the Meteor and Merian.

There is a lot of good information in this manuscript, but the presentation is of poor quality and I was left wondering why I should care about these measurements. This is a data paper but there is nothing wrong with including some science analysis to wet the reader's appetite for the dataset. Of interest is a basic analysis of the diurnal cycle, especially from the Meteor, as the ship was in a truly marine location unaffected by land and not moving around as much as the Merian. The ability to do a 24/7 analysis is a strength of these measurements I feel. Such an assessment could perhaps also shed light on whether BCO experiences an island effect. Also of interest is how the IWV/LWP vary with cloud morphology in EUREC4A - those identifications are readily available.

The presentation of the measurements themselves is also poor. A flowchart would help with discerning the subtleties between the different platforms. Fig. 1 should be a stand-alone map, a larger but more zoomed-in version of Fig. 1d, with the ship tracks superimposed on a geophysical variable of relevance such as SST or low cloud cover. The organization of the manuscript also needs further attention - the cloud mask should be described before the IWV retrievals as the cloud mask is also of relevant for the MSMRAD and LIMRAD retrievals. There was also often references to other papers (Kalesse-Los, Acquistapace) that seemed relevant but left the reader hanging.

Please give this another go. A lot of useful effort has gone into this endeavor already, but a bit more polish would make the work much more impactful.

Specific comments:

Abstract: needs to include something on the retrievals, shouldn't assume the reader knows W- K- and V- band, include the actual frequencies or wavelengths, include some basic information on where the 3 platforms were locations. A science nugget - the diurnal cycle should be a piece of cake right? - should also be included.

Introduction:
Lines 18-26: these contain some motivational platitudes, none of which convince me that these microwave measurements were really necessary. Can you come up with something more specific?
Line 27: this should start a new paragraph, needs a segue sentence.
Lines 32-38: any interesting science tidbits from these previous analyses you can mention? Would also be good to reference something from ENA and other subtropical marine campaigns, MAGIC and LASIC from DOE come to mind for me, also VOCALS.
Line 38-39: the reference to a satellite product is jarring here but you could use it as an opportunity to mention what can be done from the surface that can't be done from satellite.
Line 45: please include wavelength/frequencies for the various bands mentioned here.
Line 52: I didn't see mention anywhere as to whether ice cloud scattering could be occurring.
Lines 53-64: this paragraph is going into the instrument details, not appropriate for an introduction. This language should all go into section 2. Instead briefly describe the environmental setting EUREC4A provided for the 3 platforms and that you are evaluating two different retrievals, "HAT" and "RAD". You could also mention there was a time period with

colocated measurements, and that slight differences exist even in the treatment of identical instruments, further justifying the intercomparison. Also, one would expect from the outset that "RAD" is less precise than "HAT", no? Just because one is multi-wavelength, the other is single-wavelength. Yet the authors never say this.

Section 2:
Line 71: this would go into the introduction.
Line 73: "Tab" should be "Table" throughout.
Describe table 1 and fig 1 together.
I also think a flowchart would be nice to include here, that could be referred to as the authors go into further detail.
Personally I think you might want to begin by describing the cloud masks and how those are used. These are called 'precipitation masks' in table 1, right? Would be good to just use one nomenclature.

BCO:
Line 82: I'm confused why we need to be told the azimuth setting of a zenith measurement .
Meteor:
Line 97: what is a chirp program? And why can't we just be told what they are rather than needing to go to Kalesse-Los et al. 2023?
Lines 108-110: should go into the beginning of the section.
Merian:
Line 113-114: description of LIMRAD is out of place here.

Retrievals:
Lines 123-125: an overview of the logic would be nice here. Justify using two different retrievals. Describe the climatological training dataset. Reference a flowchart. Mention the different approaches to identifying clear-sky and why. Why do you explain the LWP retrievals before the IWV ones?
Line 133: first mention of a climatological training data set without describing what it is is a faux pas….it just means reorganizing a few sentences here.
Line 135: describe the RT model rather than telling us what other papers we need to go to to find out about it.
Line ~150: why is HATPRO being used to ID clear-sky rather than ceiloemter/radar? I've gotten confused about the different approaches and what justifies when they are applied.
Lines 154-174: you talk about how the clear-sky Tb is applied before mentioning how clear-sky is determined on the very last line (which now invokes the radar rather than HATPRO, is this because you want it to be stand-alone?)

Line 178: very unsatistifying to be told Kalesse-Los 2020 has a different LIMHAT retrieval with no explanation. Is this one better? What's different?
Line 194: why clip LWP/IWV? Do the precip masks not always work?
Line 199: don't need to be told again the BCOHAT weather station didn't work…
Line 201: are you sure reflectivity> -50dBZ indicates precip? That is an awfully low value. Does it indicate conditions that require 'clipping' of LWP/IWV? And Why -50dBZ threshold at BCO and -40dBZ at Meteor/Merian? Why not just one consistent value?

Line 215: now we learn about a radar/ceilometer cloud mask. I really think the precip/cloud masks deserve a section of their own.
Line 230: ice/cirrus is not an issue, right?

Page 11: the skewness in the IWV is rather remarkable. Did the high IWV values correspond to a particular synoptic situation? On fig. 3 it looks like the Merian maybe experienced deep tropical convective conditions around 1/26 - was it close to Brazil at the time? Worth

mentioning it's not typical of the wintertime trades if so. Which then raises the question whether the IWV distribution within the wintertime trades is genuinely log-normally distributed. I'm not sure any of the locations indicated in Fig. 2 of Foster et al. 2006 can be considered subtropical suppressed.

P.13: the ship colocation results deserve their own subsection I feel.

Line 284: doesn't ERA5 have a dry bias in the boundary layer relative to the HALO soundings? I feel I read/heard that somewhere - Geet George's paper?

Section 6.1: now we learn about the cloud mask….this section should come earlier.
Line 295: here you mention using a -50dBZ, earlier you also mentioned -40dBZ. Best to just describe in one place, correctly, and be done with it. Radar-detected sea spray is more plausible than precipitation at these low values. Doesn't Marcus Klingebiel have a paper on it? A sensitivity analysis to your choice of threshold could be good. It will mostly affect the %age of detected cloud as opposed to your LWP statistics I suspect.

Line 300: please summarize discussion in Konow et al 2021 as opposed to telling us we need to read the Konow paper to learn these things.

Line 324: I think BCO has about a half-day's advection downstream of the Meteor, over a warmer ocean with one would expect a slightly moister atmosphere. I think all you can say is that your two mean LWP values can't resolve this evolution (though is not in disagreement either).

Section 8:
Overall very disappointed to see no EUREC4A-relevant science in this manuscript, in particular, nothing on the diurnal cycle. Please do something on the diurnal cycle - it's so easy with the data you have.

Figures
Fig. 1: see top of review
Fig. 3: the red colors indicating BCO and Merian can be difficult to distinguish. Would suggest selecting a completely different color, like green.
Fig. 4: inset difficult to read, in part before of the color choices (the two reds are fairly similar)

Tables
Table 4: why so many more ERA5 soundings than radiosondes if the closest ERA5 fields are selected? Bias is calculated relative to what? Mention in caption.

---

## Author Comment (AC1)

**Answers to the Reviewer Comments**

We thank the reviewers for the constructive and thoughtful comments. Reviewers' comments are given in blue, and our respective answers are given in black. Line numbers refer to the initially submitted manuscript.

**RC 1**

This paper provides an overview of the IWV, LWP, and thermodynamic profiles retrieved from two ground-based multi-channel microwave radiometers and the passive observations from two single-frequency cloud radars. Details are provided on the retrieval methods. A detailed analysis and comparison of the IWV and LWP, including a discussion of the sensitivity of the different systems and potential sources of bias, was included. Overall, I find this paper to be well-constructed, the writing clear and precise, and that the paper should be accepted after some relatively minor suggestions are addressed. Nice job, authors!

Line 193: Would you please provide motivation to why you selected these thresholds to use in your clipping
We align the clipping thresholds with the analysis presented in Jacob et al 2019 who analyse IWV and LWP in the same region based on a neural network approach. Jacob et al 2019 argue that a LWP threshold of $1000 \mathrm{gm}^{-2}$ excludes scenes when ice scattering and precipitation impact the retrieval. Precipitation is likely present above this threshold which we exclude from our data set to avoid biases introduced due to wet radome conditions. The training data set for the single-channel retrieval (Sec. 3.2) does not include LWP above $1000 \mathrm{gm}^{-2}$; therefore, retrieved values above this threshold are not part of the conditions the retrievals was trained for.

The IWV threshold of $60 \mathrm{kgm}^{-2}$ was motivated from previous studies in the same region such as the Narval campaigns (Stevens et al 2019, BAMS: doi.org/10 .1175/ BAMS-D-18-0198.1; Jacob et al 2019, AMT: doi.org/10.5194/amt-12-3237-2019).

L193: And presumably, "clipping" means that data above these thresholds are considered "bad" and not used in future analyses (or included in the dataset you've archived)?
In the provided data set, the clipping flags are provided in Level 2 and Level 3, and corresponding IWV/LWP values are set to nan in the provided Level 4 time series. They are excluded in the analyses presented in this manuscript.  To clarify this in the manuscript, we edited L193 – 195 which now reads: „In addition to the information given by the instrument's housekeeping data, IWV values larger than $60 \mathrm{kgm}^{-2}$ , and LWP values larger than $1000 \mathrm{gm}^{-2}$ are flagged to additionally exclude poor retrieval quality or ice scattering impacts leading to errorneously high retrieval results (see e.g. Jacob et al. (2019))."

L293: I believe you mean it is clear if there is no observed backscatter above some threshold; please indicate that threshold. Also, what is the model of the ceilometer used in this study, and is the same model used on all three locations (BCO as well as the two RVs)?
We thank the reviewer for the clarifying question.  We modified the description of the cloud masking algorithm which is now available in Sec. 4.1. We modified the corresponding line in the re-submitted manuscript which now reads: „Independent cloud masking was

performed using the adjacent radar and, at BCO and aboard the Meteor, ceilometer measurements from Jenoptik/Lufft CHM15k Nimbus ceilometers. Ceilometer measurements are identified as cloudy, if a cloud base height above 100m is derived by the internal instrument software. If no valid cloud base height is derived, the scene is treated as clear. "

The same ceilometer model was used at BCO and aboard the RV Meteor. Aboard the RV Merian, no ceilometer measurements were performed. Both ceilometers operated in cloud detection mode which determines a cloud base height based on the vertical backscatter gradient. More detailed information can also be found in the manufacturer's manual (https://www.lufft.com/fileadmin/lufft.com/07_downloads/1_manuals/ Manual_CHM15k_EN_R19.pdf, Section 9 and Fig 30).

*Tab 1: Ceilometer specifications at BCO and aboard Meteor*

| site | instrument | Software version | Firmware version | Cloud detection mode |
|---|---|---|---|---|
| BCO | Jenoptik/Lufft CHM15k Nimbus | 12.12.1 (after Feb 9)

 17.05.1 (before Feb 9) | 0.747 | 0 |
| | | | | |
| Meteor | Jenoptik/Lufft CHM15k Nimbus | 17.05.1 | 0.747 | 0 |

L295: I believe you again mean "flagged clear if the cloud radar does not indicate any reflectivity above X dBZ between 500m…"
We modified the description of the cloud masking process and the respective sentence which is now available in Section 4.1.

L301: for confidently cloudy, do you require both the radar and ceilometer see the cloud at (approximately) the same height?
We currently do not demand the same height from both instruments, as we assume that any liquid cloud would be detected by the ceilometer due to its optical thickness (also see next comment).

General question: how do you identify cases where the radar sees an ice cloud, and prevent these occurrences from affecting your statistics?
We thank the reviewer for this important question. Based on this remark, we modified the cloud masking which now includes an additional flag for liquid cloud. The liquid cloud flag is True if the radar detects a reflectivity of higher than -50dBZ between 300 and 4000. Height thresholds were chosen to exclude sea spray occurrences at the lower end, and to incorporate the climatological level of the 0°-isoline.

We therefore modified the LWP analysis and statistics throughout the entire manuscript to only include liquid clouds in the statistics. A new Section 4.1 describes precipitation and cloud masking in more detail.

L347: it would be good to reference a paper here that shows that the visible, infrared, and microwave measurements have different sensitivities to liquid clouds. One possible example from the ARM program is Turner et al. BAMS 2007.

We thank the reviewer for this suggestion and added the reference to the manuscript in the corresponding sentence: „[…]  LWP retrieved from measurements by visible or infrared remote sensing techniques is not sensitive to the same column as the microwave measurements (e.g. Turner et al., 2007)"

L362: the single-channel retrievals have two aspects that result in the lower LWP noise values: (a) the IWV is fixed, and (b) the observation frequency is in the W-band, where the strength of the WV absorption is stronger. You mention the latter at L381, but it could be mentioned here too.
Thank you for this feedback. We edited the corresponding sentence which now reads: „The single-channel clear-sky LWP noises are smaller (3.9 and 4.3 $gm^{-2}$ , respectively), as IWV is fixed due to retrieving from the TB difference of cloudy and clear-sky; and as water vapor absorption is stronger in the W-band compared to the bands used in BCOHAT and LIMHAT."

L383, and also at L362: if you included realistic uncertainties in the IWV used in the single channel LWP retrieval from the radars, it would result in larger uncertainties in the retrieved LWP. But how much larger? Would you please quantify that?
We thank the reviewer for this important remark. The single-channel LWP retrieval makes use of the TB difference (TB_cloudy – TB_clear) between cloudy and closest clear-sky TB. Thus, a best estimate expressed in TB is directly included in the retrieval equation (also see Sec. 3.2 and Eq. 2).  Yet, due to the temporal difference between cloudy and clear-sky TB measurement, IWV can vary and result in an additional TB offset term.  Generally, the closest clear-sky TB measurement is available within maximal 20 minutes from each cloudy measurement due to prevailing small cloud sizes and life times. IWV varies by about 1-2kgm$^{-2}$ within a 20-minute window, resulting in a TB offset of about 3K at 89GHz. As seen in Fig 1, this additional TB offset impacts the LWP uncertainty by up to 2gm$^{-2}$ below 10gm$^{-2}$, and by approximately 0.5gm$^{-2}$ between 10 and 100gm$^{-2}$.

[Figure]

*Fig. 1: RMSD of retrieved vs true LWP for METRAD (purple) and LIMRAD (blue) single-channel retrieval with original (solid) and biased (dashed) clear-sky IWV, binned to retrieved LWP*

We added the following sentence to the manuscript: „Relative LWP uncertainty for LWP between 10 and 100gm$^{-2}$ increases by few percentage points if closest clear-sky and cloudy IWV differ by 1 to 2kgm$^{-2}$."

L429: you need to include a reference to the 2 degrees of freedom for MWR temperature retrievals. There are several papers by coauthor Löhnert that could be referenced here.
We added the following reference to the corresponding sentence:
Löhnert et al,: Ground-Based Temperature and Humidity Profiling Using Spectral Infrared and Microwave Observations. Part I: Simulated Retrieval Performance in Clear-Sky Conditions, JAMC, 48 (5), 1017–1032, doi.org/10.1175/2008JAMC2060.1, 2009

L437: Would using a physical-iterative retrieval result in improved statistics, relative to your statistical retrieval, at this site? It might be interesting to include a sentence about that (either here or in the discussion)

Great idea! Yes, indeed, a physical-iterative retrieval approach, such as Optimal Estimation (Rogers 2000), would offer an improved way of characterizing underlyingstatistics, e.g. through a resulting retrieval uncertainty, a characterization of the covariances of the prior state, and the measurement errors.

We modified the following sentence in the conclusion (L492) which now reads: „More specifically, improvements are expected by applying neural-network based (e.g. Jacob et al., 2019; Cadeddu et al., 2009) or physical (e.g. Löhnert et al, 2004; Turner et al, 2007; Maahn et al, 2020) retrieval approaches."

We also added a sentence at the end of Section 7.1 (L442): „A physical-iterative retrieval approach such as Optimal Estimation (Rogers 2000; Maahn et al, 2020) would help to constrain the covariances of the prior temperature profile data set."

Again, very nice paper.
We thank the reviewer very much for their helpful and constructive feedback!

**RC 2**

This manuscript describes the passive microwave measurements and the LWP and IWV retrievals based on them from multi-wavelength K- and V-band radiometer on the Meteor and at BCO, and single-wavelength radiometer retrievals using radar-determined cloud identification on the Meteor and Merian.

There is a lot of good information in this manuscript, but the presentation is of poor quality and I was left wondering why I should care about these measurements. This is a data paper but there is nothing wrong with including some science analysis to wet the reader's appetite for the dataset. Of interest is a basic analysis of the diurnal cycle, especially from the Meteor, as the ship was in a truly marine location unaffected by land and not moving around as much as the Merian. The ability to do a 24/7 analysis is a strength of these measurements I feel. Such an assessment could perhaps also shed light on whether BCO experiences an island effect. Also of interest is how the IWV/LWP vary with cloud morphology in EUREC4A - those identifications are readily available.

The presentation of the measurements themselves is also poor. A flowchart would help with discerning the subtleties between the different platforms. Fig. 1 should be a stand-alone map, a larger but more zoomed-in version of Fig. 1d, with the ship tracks superimposed on a geophysical variable of relevance such as SST or low cloud cover.

The organization of the manuscript also needs further attention - the cloud mask should be described before the IWV retrievals as the cloud mask is also of relevant for the MSMRAD and LIMRAD retrievals. There was also often references to other papers (Kalesse-Los, Acquistapace) that seemed relevant but left the reader hanging.
Please give this another go. A lot of useful effort has gone into this endeavor already, but a bit more polish would make the work much more impactful.

We thank the reviewer for their detailed comments which we address in detail below. More major edits on the manuscript include the description of the cloud mask (former Sec 6.1) which is now presented in its own Section 4.1 together with the precipitation masking approach.  We are currently preparing a more scientific manuscript in which the daily cycle as well as LWP conditions in different organization patterns are analyzed.

Specific comments:

Abstract: needs to include something on the retrievals, shouldn't assume the reader knows W-K- and V- band, include the actual frequencies or wavelengths, include some basic information on where the 3 platforms were locations.
We modified L2: „[…] aboard the RV Meteor and RV Maria S Merian in the downstream winter trades of the North Atlantic"
as well as L6: „Multi-channel radiometric measurements were performed at BCO and aboard the RV Meteor between 22 and 31 GHz (K-band), and 51 to 58 GHz (V-band). Combined […]"
and added a sentence on the retrieval method (L9):"We present a novel retrieval method to retrieve LWP from single-channel 89GHz measurements, evaluate retrieved quantities with independent measurements and analyze retrieval uncertainties e.g. by site inter-comparison."

A science nugget - the diurnal cycle should be a piece of cake right? - should also be included.
We refer the reviewer to a future publication with a scientific focus that is currently in preparation.

Introduction:
Lines 18-26: these contain some motivational platitudes, none of which convince me that these microwave measurements were really necessary. Can you come up with something more specific?
We added the following sentence to the introduction: "These measurements manifest an important contribution to the overall data set as they quantify cloud liquid water amount statistically at high temporal resolution."

Line 27: this should start a new paragraph, needs a segue sentence.
We added a line break before the sentence „In order to elucidate [...]"

Lines 32-38: any interesting science tidbits from these previous analyses you can mention? Would also be good to reference something from ENA and other subtropical marine campaigns, MAGIC and LASIC from DOE come to mind for me, also VOCALS.
We thank the reviewer for this suggestion. Literature suggests that MAGIC (Zhou et al, 2015, BAMS), LASIC (Zuidema et al, 2016, BAMS) and VOCALS-ReX (Mechoso et al, 2015, BAMS) rather targeted Stratocumulus clouds and their transitions. However, the RICO campaign targeted Cumulus convection in the same region. We therefore added the following sentences to the manuscript: „LWP conditions in the Northern Atlantic winter trades have been previously measured  during the RICO (Rain in shallow Cumulus over the Ocean, Rauber et al, 2005) campaign."

We additionally edited the following paragraph to incorporate some of previous scientific findings and the importance of the presented data set for satellite product evaluation and model benchmarking.

Line 38-39: the reference to a satellite product is jarring here but you could use it as an opportunity to mention what can be done from the surface that can't be done from satellite.
See above

Line 45: please include wavelength/frequencies for the various bands mentioned here.
The sentence now reads: „While water vapor and oxygen emit in distinct absorption bands in the K- and G-band (around 22.2 GHz and 183.3 GHz, respectively), and V- and F-band (60.0 GHz and 118.8 GHz, respectively), liquid water emissions increase with increasing frequency (Ulaby, 2014)."

Line 52: I didn't see mention anywhere as to whether ice cloud scattering could be occurring.
We modified the cloud masking approach as summarized in a new Section 4.1 to include a liquid cloud flag (see RC 1).

Lines 53-64: this paragraph is going into the instrument details, not appropriate for an introduction. This language should all go into section 2. Instead briefly describe the environmental setting EUREC4A provided for the 3 platforms and that you are evaluating

two different retrievals, "HAT" and "RAD". You could also mention there was a time period
with colocated measurements, and that slight differences exist even in the treatment of
identical instruments, further justifying the intercomparison. Also, one would expect from
the outset that "RAD" is less precise than "HAT", no? Just because one is multi-
wavelength, the other is single-wavelength. Yet the authors never say this.
We thank the reviewer for the suggestions and greatly modified the corresponding
paragraph.

Section 2:

Line 71: this would go into the introduction.
We think that the corresponding statement in L29-30 is sufficient.

Line 73: "Tab" should be "Table" throughout.
We changed this throughout the entire manuscript.

Describe table 1 and fig 1 together. I also think a flowchart would be nice to include here,
that could be referred to as the authors go into further detail. Personally I think you might
want to begin by describing the cloud masks and how those are used. These are called
'precipitation masks' in table 1, right? Would be good to just use one nomenclature.
We thank the reviewer for this comment and removed the precipitation masks from Tab 1
to avoid confusion. Cloud and precipitation masking is now explained in a new Section 4.1.

BCO:
 Line 82: I'm confused why we need to be told the azimuth setting of a zenith
measurement.
We thank the reviewer for this remark and modified the respective line which now reads:
„Zenith measurements are performed for 15 minutes at a temporal resolution of 2
seconds".

Meteor:
Line 97: what is a chirp program? And why can't we just be told what they are rather than
needing to go to Kalesse-Los et al. 2023?
Frequency-modulated-continuous-wave (FMCW) radar continuously emit sawtooth chirp
sequences over a certain bandwidth around the center frequency (here: 94 GHz). Due to
technical constraints, one radar chirp sequence of the used RPG-FMCW-94-DP cannot
cover the entire unambiguous range (max. 18 km). Therefore, several chirp sequences
(also known as chirp programs or chirp tables) are executed consecutively to sample the
entire troposphere. More details are given in the instrument manual
(https://www.radiometer-physics.de/downloadftp/pub/PDF/Cloud%20Radar/RPG-FMCW-
Instrument_Manual.pdf) and Küchler et al., 2017 (https://doi.org/10.1175/JTECH-D-17-
0019.1).

Specifications of each chirp can be chosen by the user and we had decided to operate the
radar in two different settings (i.e. using two different chirp programs). The entire list of
varied radar setting parameters is rather extensive and the readers are instead pointed to
Kalesse-Los et al., 2023. Table 2 from this reference is included here:

**Table 2.** Specifications and program settings for LIMRAD94. Two main chirp tables with slightly different settings were used during the campaign. The upper row denotes the first chirp table operated from 17 to 29 January 2020 18:00 UTC, and the second row refers to the second chirp table operated from 31 January 2020 22:28 UTC to 29 February 2020 (here data until 19 February 2020 obtained in the EUREC$^4$A region of interest were used).

| Attributes | Chirp sequence 1 | Chirp sequence 2 | Chirp sequence 3 |
|---|---|---|---|
| Integration time (s) | 1.022 | 0.947 | 0.966 |
| | 0.563 | 0.573 | 0.453 |
| Range interval (m) | 300–3600 | 3600–8000 | 8000–15 000 |
| | 300–3000 | 3000–6200 | 6200–13 000 |
| Range vertical resolution (m) | 22.4 | 25.6 | 29.8 |
| | 22.4 | 37.7 | 42.1 |
| Nyquist velocity (m s$^{-1}$) | 6.4 | 5.2 | 2.9 |
| | 7.3 | 6.1 | 4.5 |
| Doppler velocity resolution (m s$^{-1}$) | 0.050 | 0.081 | 0.089 |
| | 0.057 | 0.095 | 0.070 |
| Doppler velocity bins | 256 | 128 | 64 |
| | 256 | 128 | 128 |

To increase readability, we however changed the wording on line 97 to "two different radar settings".

Lines 108-110: should go into the beginning of the section.
We moved the corresponding paragraph to the introductory paragraph of Sec 2 (L74).

Merian:
Line 113-114: description of LIMRAD is out of place here.
We clarified the respective sentence which now reads: „Aboard the Merian, the Institute for Geophysics and Meteorology of the University of Cologne operated a radar-radiometer system of the type RPG-FMCW-94 dual polarization (DP) which measures in the W-band (94 GHz) and includes a passive radiometer channel at 89 GHz (Küchler et al., 2017, here referred to as MSMRAD). MSMRAD is of the same type as LIMRAD."

Retrievals:
Lines 123-125: an overview of the logic would be nice here. Justify using two different retrievals. Describe the climatological training dataset. Reference a flowchart. Mention the different approaches to identifying clear-sky and why. Why do you explain the LWP retrievals before the IWV ones?
Based on the reviewer's comment, we added further explanation on the different retrieval methods in the introduction to Section 3, and re-organized some of the text to clarify the methods used.

Line 133: first mention of a climatological training data set without describing what it is is a faux pas....it just means reorganizing a few sentences here.
We re-organized the corresponding paragraph.

Line 135: describe the RT model rather than telling us what other papers we need to go to to find out about it.
We added a description of the RT model in the revised manuscript (L156-159).

As the retrieval is designed to work on the Hatpro measurements stand-alone, this additional LWP retrieval correction scheme is applied as explained. The ceilometer/radar measurements are used for retrieval evaluation and quantification of LWP retrieval uncertainty as analyzed in Sec. 6.2.

Lines 154-174: you talk about how the clear-sky Tb is applied before mentioning how clear-sky is determined on the very last line (which now invokes the radar rather than HATPRO, is this because you want it to be stand-alone?)
Motivated from the reviewer's question, we modified the description of the retrieval to clarify the approach.

Line 178: very unsatistifying to be told Kalesse-Los 2020 has a different LIMHAT retrieval with no explanation. Is this one better? What's different?
We thank the reviewer for this question. We actually were mistaken and the same retrieval was used in both data sets. Therefore, we deleted the corresponding sentence, and added the references to the level 1 data sets as follows (L178): „The LIMHAT data set is also available in Kalesse-Los et al, 2020. „

Line 194: why clip LWP/IWV? Do the precip masks not always work?
Please be referred to the first comment of RC 1.

Line 199: don't need to be told again the BCOHAT weather station didn't work...
The description on the precipitation masks was moved to Sec. 3.3

Line 201: are you sure reflectivity> -50dBZ indicates precip? That is an awfully low value. Does it indicate conditions that require 'clipping' of LWP/IWV?
At 250m above ground, we would normally expect any measured reflectivity signal to be due to precipitation. Yet, at BCO, sea spray clutter is visible in the radar as analyzed in Klingebiel et al, 2019 in reflectivity range between -50 and -65dBZ. By applying a reflectivity threshold of >-50dBZ to the lowest range gates in the precipitation flagging, we therefore exclude sea-spray occurrences from being flagged as precipitating.

And Why -50dBZ threshold at BCO and -40dBZ at Meteor/Merian? Why not just one consistent value?
Two different sensitivity thresholds were applied due to the different sensitivities of the two radars following from different chirp table settings (see L203; and above).

Line 215: now we learn about a radar/ceilometer cloud mask. I really think the precip/cloud masks deserve a section of their own.
We thank the reviewer for the suggestion. We added Section 4.1 to introduce the different precipitation and cloud masking tools, and adjusted respective parts of Sec. 4 and 6 accordingly.

Line 230: ice/cirrus is not an issue, right?
We thank the reviewer for the comment.  By modifying the cloud masking approach (see previous comments and RC1), we only take into account liquid clouds in all LWP analyses.

Page 11: the skewness in the IWV is rather remarkable. Did the high IWV values correspond to a particular synoptic situation? On fig. 3 it looks like the Merian maybe

experienced deep tropical convective conditions around 1/26 - was it close to Brazil at the time? Worth mentioning it's not typical of the wintertime trades if so.
Indeed, the high IWV in the distribution corresponds to the deep convective features that the Merian experienced around Jan 26. We added the following lines to the manuscript (L239): „Higher IWV conditions of more than 50 kgm$^{-2}$, untypical for winter trade conditions, were observed close to Brazil from January 27 until 29, 2020, associated with a deep convective system. „

Which then raises the question whether the IWV distribution within the wintertime trades is genuinely log-normally distributed. I'm not sure any of the locations indicated in Fig. 2 of Foster et al. 2006 can be considered subtropical suppressed.
We thank the reviewer for this interesting comment. Future analyses making use of the almost 10-year long climatology of HATPRO IWV measurements at BCO will investigate this question which is out of scope of this data publication.

P.13: the ship colocation results deserve their own subsection I feel.
We added subsection 6.3 to the manuscript to incorporate the reviewer's suggestion.

Line 284: doesn't ERA5 have a dry bias in the boundary layer relative to the HALO soundings? I feel I read/heard that somewhere - Geet George's paper?
We assume that the reviewer is referring to George et al 2021 (ESSD, doi.org/10.5194/essd-13-5253-2021). The authors find that the HALO dropsondes experience a dry bias compared to the radiosounding data set (available in Stephan et al, 2020, ESSD) which they correct through a multiplicative correction factor (see Sec. 4 in their paper) in higher data levels. As we do not make use of the dropsonde data set, this dry bias should not impact our analyses.

Section 6.1: now we learn about the cloud mask....this section should come earlier.
We thank the reviewer for this suggestion and implemented a new Section 4.1 describing the precipitation and cloud masking.

Line 295: here you mention using a -50dBZ, earlier you also mentioned -40dBZ. Best to just describe in one place, correctly, and be done with it.
See above and in new Section 4.1

Radar-detected sea spray is more plausible than precipitation at these low values. Doesn't Marcus Klingebiel have a paper on it?
We refer the reviewer to the answer of the comment targeting L201 further above, and included the reference in the modified precipitation mask description in Sec. 4.1.

A sensitivity analysis to your choice of threshold could be good. It will mostly affect the %age of detected cloud as opposed to your LWP statistics I suspect.
We thank the reviewer for this important remark. We ran the cloud masking algorithm with a radar reflectivity threshold of -45dBZ at all sites to indicate cloudiness as opposed to -50dBZ (BCO, Merian) and -40dBZ (Meteor), respectively. Tab. 2 summarizes the resulting changes in confident liquid cloudy cloud fraction, as well as mean LWP. Changes in mean LWP are within the associated LWP uncertainties. Fig. 2 shows that the threshold only marginally impacts the distribution.

*Tab 2: Sensitivity of confident liquid cloud cloudiness and mean LWP to radar reflectivity threshold in cloud mask algorithm*

| instrument | Initial CM cloudiness / % | Modified CM cloudiness / % | Initial mean LWP / gm$^{-2}$ | Modified CM mean LWP / gm$^{-2}$ |
|---|---|---|---|---|
| BCOHAT | 33.5 | 32.7 | 64.8 | 68.5 |
| LIMHAT | 19.3 | 19.9 | 62.5 | 60.8 |
| LIMRAD | 21.5 | 22.1 | 52.4 | 50.9 |
| MSMRAD | 21.0 | 21.0 | 46.8 | 46.8 |

[Figure]

*Fig 2: Difference of distributions of LWP with original and modified cloud masking reflectivity threshold for BCOHAT (red), LIMHAT (blue), LIMRAD (cyan), MSMRAD (purple)*

Line 300: please summarize discussion in Konow et al 2021 as opposed to telling us we need to read the Konow paper to learn these things.
L300 summarizes the main factors that Konow et al (2021) find to explain the occurrence of probably cloudy in their airborne data set. To clarify the sentence, we changed the wording of L300 which now reads: „[…] between the ceilometer and radar as outlined in Konow et al (2021)."

Line 324: I think BCO has about a half-day's advection downstream of the Meteor, over a warmer ocean with one would expect a slightly moister atmosphere. I think all you can say is that your two mean LWP values can't resolve this evolution (though is not in disagreement either).

We thank the reviewer for this remark and modified the corresponding sentence which now reads: „Even though similar mean LWP conditions were observed, more detailed trajectory analyses are necessary to investigate the effect of ocean surface and island impact on the cloud evolution between Meteor and BCO.“

Section 8:
Overall very disappointed to see no EUREC4A-relevant science in this manuscript, in particular, nothing on the diurnal cycle. Please do something on the diurnal cycle - it's so easy with the data you have.
Thank you for the suggestion. Current analyses are looking at that, and will be published in a designated publication. We adjusted L495 which now reads: „The spatial dimension of this data set are currently further exploited by characterizing LWP and IWV conditions in different mesoscale organization conditions (e.g. Schulz et al., 2021), and by evaluating microwave and vis/IR satellite LWP products as well as climatologies (Elsaesser et al., 2017).“

Figures
Fig. 1: see top of review
Larger maps of the EUREC4A operations are available in the literature: Stevens et al2021 Fig 4; Acquistapace et al 2022 Fig 2. In order to keep the manuscript at reasonable length, we would like to keep the figure as it is.

Fig. 3: the red colors indicating BCO and Merian can be difficult to distinguish. Would suggest selecting a completely different color, like green.
We modified the color scheme. Throughout the manuscript, BCO is given in red, Meteor LIMHAT in blue, Meteor LIMRAD in cyan, Merian MSMRAD in violet.

Fig. 4: inset difficult to read, in part before of the color choices (the two reds are fairly similar)
see above

Tables
Table 4: why so many more ERA5 soundings than radiosondes if the closest ERA5 fields are selected?
The number N provided in the ERA5 column refers to the number of available closest ERA5 fields and can be understood as number of available measurement points with availble ERA5 and MWR measurement. This number is independent from the number of radiosoundings.

Bias is calculated relative to what? Mention in caption.
We added the following sentence to the caption: „A positive bias refers to drier MWR conditions than measured by the respective independent IWV measurement.“